# Sketchy: Memory-efficient Adaptive Regularization with Frequent Directions

**Vladimir Feinberg**
Google DeepMind
vladf@google.com

**Xinyi Chen**
Princeton University
Google DeepMind

**Y. Jennifer Sun**
Princeton University
Google DeepMind

**Rohan Anil**
Google DeepMind

**Elad Hazan**
Princeton University
Google DeepMind

## Abstract

Adaptive regularization methods that exploit more than the diagonal entries exhibit state of the art performance for many tasks, but can be prohibitive in terms of memory and running time. We find the spectra of the Kronecker-factored gradient covariance matrix in deep learning (DL) training tasks are concentrated on a small leading eigenspace that changes throughout training, motivating a low-rank sketching approach. We describe a generic method for reducing memory and compute requirements of maintaining a matrix preconditioner using the Frequent Directions (FD) sketch. While previous approaches have explored applying FD for second-order optimization, we present a novel analysis which allows efficient interpolation between resource requirements and the degradation in regret guarantees with rank $k$: in the online convex optimization (OCO) setting over dimension $d$, we match full-matrix $d^2$ memory regret using only $dk$ memory up to additive error in the bottom $d - k$ eigenvalues of the gradient covariance. Further, we show extensions of our work to Shampoo, resulting in a method competitive in quality with Shampoo and Adam, yet requiring only sub-linear memory for tracking second moments.

## 1 Introduction

DL optimization commonly relies on adaptive gradient methods, namely the Adam optimizer [1]. It differs from stochastic gradient descent in that the learning rate is a structured diagonal matrix built from previous gradients rather than a scalar. In full matrix AdaGrad [2], the inverse matrix square root of the sum of outer products of previous gradients is the learning rate.

Full matrix preconditioning is impractical for modern deep learning architectures: for instance, the ResNet-50 architecture [3] has over 23 million parameters, requiring more than 2 petabytes to represent its gradient covariance. Thus, diagonal preconditioning methods remain popular. However, previous work has demonstrated state-of-the-art results in some settings, such as large-batch data parallel training, for nondiagonal forms of preconditioning [4, 5, 6, 7, 8, 9]. In particular, Shampoo [5, 9] introduces a factorization of full matrix preconditioning method with adoption in large-scale industrial applications such as training Google's ads click-through-rate model [10]. Furthermore, as hardware evolves, memory efficiency becomes an increasing concern, as "logic improves much faster than wires and SRAM, so logic is relatively free" [11]: from TPUv2 to TPUv3, per-chip `bfloat16` operations per second improved $2.67\times$ but memory bandwidth only improved $1.29\times$. GPUs exhibit a similar pattern for compute and memory increase, at $5\times$ and $2.2\times$, for V100 to A100 [12].

37th Conference on Neural Information Processing Systems (NeurIPS 2023).

Investigation into the Kronecker-factored gradient covariance matrix reveals a concentrated, but changing, spectrum (Fig. 3), suggesting the majority of the spectral mass can be represented by a low-rank matrix, albeit rotating over time. The Frequent Directions (FD) sketch provides a mechanism to track the top eigenvectors without materializing the full covariance matrix, as proposed in [13]. Is a large portion of the spectral mass sufficient to retain the performance of adaptive regularization in theory and practice? In this work, we investigate this hypothesis.

- In the setting of online convex optimization, by applying a dynamic diagonal regularization to the FD sketch, we can **recover full-matrix AdaGrad regret up to additive spectral terms under a memory constraint**, providing a novel guarantee without curvature assumptions (Sec. 4.1). Rigorously composing our approach with Shampoo (Sec. 4.2) unlocks a **second-order algorithm which requires sub-linear memory** for its accumulators.

- By modifying FD for exponential moving averages (Sec. 4.3), we demonstrate a practical algorithm competitive with at-least-linear memory Shampoo and Adam in three modern DL settings (Sec. 5.1). While previous work [8, 14] shows rank-1 preconditioners are effective for trading off quality for memory, these results **demonstrate a Pareto improvement by using higher-rank approximations**.

- We explain the competitive performance of Sketchy with observations of fast spectral decay in the moving average of Kronecker-factored gradient covariance in DL settings (Sec. 5.2).

## 2    Setting and Definitions

**Regret and Optimization.**    The optimization problem of training a deep neural network has a non-convex objective loss function $f$. Since finding the global optimum is computationally intractable in general, theoretical guarantees focus on convergence to an $\varepsilon$-approximate first-order optimum: a point $x$ such that $\|\nabla f(x)\| \leq \varepsilon$. A smooth non-convex problem can be reduced to solving a series of offline convex problems [6]. The convex sub-problems have form $f_t(x) = f(x) + c\|x - x_t\|^2$, where $c$ is a constant and $x_t$ is an iterate in the optimization process. Using online-to-batch conversion, we can translate the regret bound of an online convex optimization (OCO) [15] algorithm to convergence guarantees for offline optimization. For more details of this reduction, see Appendix B.1. Therefore, non-convex optimization guarantees can be obtained from regret bounds, and we focus on the latter in this paper.

In this setting, an OCO algorithm chooses a point $x_t \in \mathcal{K}$ iteratively, where $\mathcal{K} \subseteq \mathbb{R}^d$ is a convex decision set (take $\mathcal{K} = \mathbb{R}^d$ if unconstrained). After the decision is made, the adversary reveals a convex loss function $f_t$, to which the algorithm suffers cost $f_t(x_t)$. Upon receiving the cost, the algorithm updates its decision for the next iteration. The regret for an online algorithm is given by

$$\text{Regret}_T = \sum_{t=1}^{T} f_t(x_t) - \min_{x \in \mathcal{K}} \sum_{t=1}^{T} f_t(x) \ .$$

**Sketching and the Frequent Directions Method.**    Given a stream of vectors $g_t \in \mathbb{R}^d$, $t \in [T]$, we utilize the FD sketch [13] given in Alg. 1 which maintains a low-rank approximation of the true running covariance $G_t = \sum_{s \leq t} g_s g_s^\top$. At each time $t$, it maintains a matrix $B_t$ of size $d \times \ell$ whose last column is 0 and whose square is the sketch $B_t B_t^\top = \bar{G}_t$. After seeing $g_t$ from the stream, we update the previous matrix using Alg. 1, which updates $B_{t+1}$ of size $d \times \ell$ whose last column remains 0; take $B_0 = 0$. $B_{t+1}$ is obtained by decomposing the sum of $\bar{G}_t$ and the newly observed matrix, keeping only the top eigendirections, and reducing the eigenvalues uniformly by $\ell$-th eigenvalue. At every iteration $t$, denote $\rho_t := \lambda_\ell^{(t)}$ be the removed eigenvalue from the covariance update in Alg. 1.

For convenience, let $\rho_{1:t} \overset{\text{def}}{=} \sum_{s=1}^{t} \rho_s$ be the cumulative escaped mass. For a matrix $X$, we denote its $i$-th leading eigenvalue by $\lambda_i(X)$. Let $\|\cdot\|_F$ denote the Frobenius norm of a matrix.

The fundamental property of FD is that applying Alg. 1 over a stream of vectors $g_t$, with $B_0 = 0$, the sum of escaped mass $\rho_t = \lambda_\ell^{(t)}$ can be bounded by the bottom eigenvalues of $G_T$, formally given by the following lemma:

---

**Algorithm 1** Frequent Directions Update (`FD-update`)

---

**Require:** Invariant that last column of $B_{t-1}$ is 0.
**Ensure:** The last column of $B_t$ is 0.
 1: Input: Previous state $\bar{G}_{t-1} = B_{t-1}B_{t-1}^\top \in \mathbb{R}^{d \times d}$
 2: Input: New symmetric PSD matrix $M_t \in \mathbb{R}^{d \times d}$.
 3: Eigendecompose $\bar{U}_t \operatorname{diag} \lambda^{(t)} \bar{U}_t^\top = \bar{G}_{t-1} + M_t$ where $\lambda^{(t)}$ contains descending eigenvalues.
 4: Define $U_t$ as the matrix whose columns are the first $\ell$ columns of $\bar{U}_t$, and $\lambda^{(t)}_{[1:\ell]}$ be its eigenvalues.
 5: Update $B_t = U_t \operatorname{diag} \left( \lambda^{(t)}_{[1:\ell]} - \lambda^{(t)}_\ell \right)^{1/2}$.
**output** $\lambda^{(t)}_\ell, B_t B_t^\top$.

---

**Lemma 1** (Liberty [16]). *The cumulative escaped mass $\rho_{1:T}$ can be upper bounded as*

$$\rho_{1:T} \leq \min_{k=0,\dots,\ell-1} \frac{\sum_{i=k+1}^d \lambda_i(G_T)}{\ell - k} \leq \sum_{i=\ell}^d \lambda_i(G_T) \stackrel{def}{=} \lambda_{\ell:d} \ .$$

*Proof.* See Sec. B.2 □

## 3 Related Work

### 3.1 Spectral Analysis of DL Training

Denote the loss function of the $i$-th example for weights $x$ as $f_i(x)$. The spectrum of the Hessian matrix $\sum_i \nabla^2 f_i$ has been the subject of intensive investigation in DL [17, 18, 19, 20] and its properties have been used to devise training methods [4, 21].

Recent papers [22, 23, 24] inspect the covariance matrix, $\sum_i (\nabla f_i)(\nabla f_i)^\top$. In small models, where its computation is feasible, these works identify fast spectral decay.

Agarwal et al. [6] take advantage of this observation by using a low-rank approximation of the whole covariance matrix, based on a limited history of the gradients, $\sum_{i=t-r}^r (\nabla f_i)(\nabla f_i)^\top$. This approach still requires $r$ copies of the model gradients in memory, where typically $r$ should scale with $\beta_2^{-1}$, with $\beta_2$ the exponential moving average for second order statistics (the authors set $r = 200$). Fundamentally, approximating the whole covariance matrix constrains Agarwal et al. [6] application to small models.

In our work, we validate the decay hypothesis holds across the per-layer factored covariance matrices in several modern neural networks. For a layer's gradient matrix $G_i$ at the $i$-th example and a second moment decay term $\beta_2$, our work inspects spectral decay for $L_t = \sum_i \beta_2^{t-i} G_i G_i^\top$ and $R_t = \sum_i \beta_2^{t-i} G_i^\top G_i$; the spectral structure for these outer products is not well-documented. Furthermore, as described in Sec. 3.4, approximating the factored covariance $L_t \otimes R_t$ requires less memory than the full covariance and explains why our method can scale to large modern architectures whereas Agarwal et al. [6] cannot.

### 3.2 Sublinear Memory Methods

Extreme Tensoring [7], AdaFactor [14], and SM3 [8] are methods that require sublinear memory relative to the number of parameters, at the other end of the memory-quality tradeoff beyond methods that rely on the diagonal of the gradient covariance such as Adam. Owing to different structural assumptions on the set of feasible preconditioners, comparison with these methods is out of scope. However, these methods may compose with our approach. One may apply Extreme Tensoring first, then sketch the resulting reshaped tensor covariances with our method to further reduce memory consumption. Similarly, an SM3-like approach which reduces the indices for each dimension to be preconditioned can be applied before Sketchy is applied to remaining indices.

Crucially, Adam, which uses linear memory for second moment representations, compares favorably in terms of quality to all of these sublinear-memory methods. **By increasing rank in factored**

Table 1: Memory-efficient adaptive gradient methods, in the OCO setting with dimension $d$ (Sec. 2). We describe the worst-case regret bounds without exp-concavity assumptions, asymptotically, hiding logarithmic factors, treating the decision set diameter as a constant, and assume optimally-tuned hyper-parameters. $\ell$ refers to the controllable preconditioner rank. Note $\operatorname{tr} G_T^{1/2} = \sqrt{\min_{H \in \mathcal{H}} \sum_t \|\nabla_t\|_H^2}$ is the optimal preconditioner's regret among the class of positive semi-definite, unit-trace matrices, $\mathcal{H}$, and $G_T$ is the sum of gradient outer products. We let eigenvalues $\lambda_i = \lambda_i(G_T)$ with $\lambda_{i:j} = \sum_{m=i}^{j} \lambda_m$.

| Reference | Regret (general convex) | Memory |
|---|---|---|
| Full Matrix AdaGrad [2] | $\operatorname{tr} G_T^{1/2}$ | $d^2$ |
| Ada-LR [25] | $\operatorname{tr} G_T^{1/2} + \lambda_{\ell+1}^{1/2} \ell^{3/4} d^{1/4}$ | $d^2$ |
| Ada-FD [26] | $\Omega\left(T^{3/4}\right)$ [1] | $d\ell$ |
| SON [27] | $\sqrt{Td}$ | $d^2$ |
| FD-SON [27] | $\sqrt{\ell \lambda_{\ell:d} T}$ | $d\ell$ |
| This paper | $\operatorname{tr}(G_T^{1/2}) + \sqrt{d(d-\ell)\lambda_{\ell:d}}$ | $d\ell$ |

**covariance representation, Sketchy is competitive with Adam, despite sublinear memory for second moment representation.** Thus, for simplicity, we compare to only Adam, which dominates the alternative sublinear approaches in terms of quality.

## 3.3 Sketching-based Approaches

Several works have explored sketching-like approximations to the gradient covariance matrix, but none provide an adaptive bound exploiting fast spectral decay in gradient covariance without additional assumptions (Tbl. 1). In this section, we consider the OCO setting over dimension $d$ (Sec. 2).

Random projection (Ada-LR) is most spiritually similar to our work [25]. Although it does not reduce memory usage, it relies on random projections to lower dimension $\ell \leq d$ to reduce inverse matrix computation costs. An alternative without formal guarantees, RadaGrad, reduces memory consumption to $O(d\ell)$; however, as with all Johnson-Lindenstraus projection methods, it suffers a probabilistic failure rate scaling as $O(\ell^{-1})$ (in comparison, our method inherits FD's determinism).

Frequent Directions (FD) [13, 16], provides an alternative matrix sketching approach from the data streaming literature. As an adaptive sketch, it dominates random projection in terms of matrix recovery, and lower bounds show its memory usage is optimal in the sense that any equally-accurate approximation to an adversarially-chosen true covariance $G_T$ in operator norm constructed from the corresponding gradients must use $O(d\ell)$ bits.

In the context of exp-concave cost functions, Luo et al. [27] provide an FD sketched version of Online Newton Step (ONS), FD-SON. In this setting, their approach nearly recovers classical ONS regret, up to logarithmic error in $\sum_{i=1}^{\ell-1} \lambda_i(G_T)$ and additive error in $\sum_{i=\ell}^{d} \lambda_i(G_T)$. However, without the exp-concave assumption, FD-SON falls back to a gradient-descent-like default regret of $O\left(\lambda_{\ell:d} \sqrt{T}\right)$, which can be $\Omega(T)$ without spectral decay. In the context of linear bandits, Chen et al. [28] uses FD for memory reduction. The resulting algorithm, SCFD, is similar to Alg. 2, but SCFD lacks a projection step and does not handle general domains $\mathcal{K}$. We emphasize that our contribution is in the novelty of our regret analysis for general OCO settings, in contrast to linear bandits.

The main prior work exploring FD for general online convex settings, Wan and Zhang [26], extends the FD-SON approach by adding a fixed diagonal perturbation $\delta I$ to an FD-based preconditioner, in Ada-FD. However, this approach does not achieve $\sqrt{T}$ regret even in a non-adversarial setting with stochastic linear cost functions (Observation 2), where learning rate and $\delta$ are tuned. Dynamically changing diagonal regularization is essential for worst-case $O\left(\sqrt{T}\right)$ performance.

**Observation 2.** *Suppose we receive linear cost functions $f_t(x) = \langle x, g_t \rangle$, where $g_t \in \mathbb{R}^d$ is a random vector drawn iid from any distribution over $r \leq d$ orthonormal vectors $W$. For any sketch size $\ell \leq r$, the bound on the expected regret of Ada-FD is $\Omega(T^{3/4})$.*

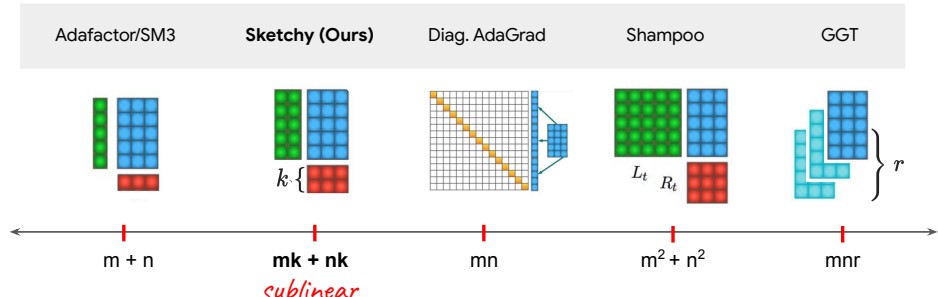

Figure 1: Asymptotic memory consumption for representing gradient covariance in adaptive regularization approaches for a single matrix parameter of size $n \times m$. Here, $r$ refers to the GGT history buffer size and $k$ to the approximation rank of FD (both typically set to hundreds). Past sketching approaches like Ada-FD and Radagrad take memory similar to GGT, with $r$ being sketch size; these are all asymptotically superlinear. This figure demonstrates optimizer memory usage in the theoretical OCO setting; in practice for deep learning workloads there are additive $O(mn)$ factors for momentum, the parameters themselves, and grafting parameters for Shampoo and Sketchy.

*Proof.* See Sec. B.3 □

Wan and Zhang [26] remark Ada-FD has $\sqrt{T}$ regret when $G_T$ is low rank with rank below $k$, so $d - k$ of its eigenvalues are precisely zero. However, this setting does not require any sketching in the first place. By tracking the column space of observed gradients (e.g., with a reduced QR decomposition, rank-1-updated every step), the full matrix AdaGrad algorithm can be perfectly recovered without using more than $O(dk)$ memory.

In concrete convex examples, Sketchy compares favorably to these approaches (Appendix A).

### 3.4 Shampoo

Perhaps our most compelling application is reducing the memory of Shampoo [5, 9, 10]. Shampoo is an adaptive preconditioning method that takes into account the structure of the parameter space, and thus is more efficient than full matrix AdaGrad. For example, if the parameter is a weight matrix $W$ of size $m \times n$, AdaGrad treats the matrix-shaped parameters as a vector of size $mn$, and the preconditioner has size $m^2 n^2$; Shampoo instead has left and right preconditioners $L, R$ of size $n \times n$ and $m \times m$, respectively, with the preconditioned update $L^{-1/4} W R^{-1/4}$. Write $\overline{\text{vec}}(W)$ as the vectorized weights, then it is equivalent to $(L \otimes R)\overline{\text{vec}}(W) = \overline{\text{vec}}(LWR)$, where $\otimes$ denotes the Kronecker product. Figure 1 illustrates the updates of AdaGrad and Shampoo, where AdaGrad update uses the entire matrix instead of the diagonal, which is shown in the figure. In other words, Shampoo uses a Kronecker-factored preconditioner, and the factorization preserves the matrix structure of the parameters. Since in DL optimization parameters often have matrix structure, Shampoo has strong empirical performance, and has improved upon state-of-the-art results in large-scale tasks such as language modeling with BERT-Large and image classification on ImageNet in [9].

Figure 1 elaborates why the composition of FD and Shampoo is essential to avoid memory consumption asymptotically greater than parameter count for approximate full matrix regularization.

However, Shampoo memory costs may still be prohibitive for rectangular weight matrices. In BERT-Large [29], most parameters are in the feed-forward network layers, which consist of $4096 \times 1024$ dense kernels; other transformers follow similar narrow-to-wide patterns. For large models, occupying even $4\times$ memory for the left preconditioner can frequently result in OOM in memory-constrained settings; this was in fact one of the practical motivations for our proposed approach.

Anil et al. [9] introduces two workarounds for the problem of rectangular matrices based on limiting covariance modelling. Furthermore, both approximations can be applied to our method, so we do not compare against them. First, the authors propose Blocked Shampoo, which views each weight matrix $W$ of shape $m \times n$ as $mn/b^2$ blocks of size $b \times b$ for some block size $b < \min(m, n)$ (in the limit $b = 1$, this recovers diagonal AdaGrad). This approach is dependent on the ordering of neurons in hidden layers. Another approximation relies on only one-sided covariance upper bounds, $L_t \otimes I$ or $I \otimes R_t$. Note, however, that the one-sided approximation doesn't help with vector parameters, such as

those that appear for the bias terms in dense layers or layer norms [30]. For 3D weights, such as those which appear in homogeneous Mixtures of Experts [31], blocking increases memory consumption. These approaches do not take into account the fast decay of the preconditioner's spectrum, which is the focus of our work.

# 4 Algorithms and Main Theorems

In this section, we introduce the adaptation of Frequent Directions (FD) to AdaGrad (Sec. 4.1) and Shampoo (Sec. 4.2), the corresponding algorithms and regret guarantees. Additionally, in Sec. 4.3, we modify FD to support exponential moving averages.

The main technical novelty in incorporating Alg. 1 to AdaGrad (Alg. 2) and Shampoo (Alg. 3) is the construction of preconditioning matrices with FD-sketched matrices compensated by the cumulative escaped masses. The insight of such construction lies in the observation that while the FD sketch lower bounds the full preconditioning matrix, the FD sketch compensated with the cumulative escaped masses upper bounds the full preconditioning matrix, as demonstrated in Lemma 10 for AdaGrad and Lemma 14 for Shampoo. The regret guarantee for AdaGrad ([2]) and Shampoo ([5]) directly depends on the trace of the preconditioning matrices, therefore obtaining upper and lower bounds on the preconditioning matrices allows explicit additive dependence on the cumulative escaped mass. We expect this approach to be reusable for alternative approximation schemes.

## 4.1 FD for AdaGrad

Our main algorithm in this section is Alg. 2 run with FD (Alg. 1) as the sketching method. $\tilde{G}_t^{-1/2}$ in Alg. 2 denotes the Moore-Penrose pseudoinverse of the matrix $\tilde{G}_t^{1/2}$. Our main algorithm, Sketchy AdaGrad, in this section exploits the FD approach outlined in Alg. 1 as the sketching method in AdaGrad. In particular, at every time step, we pass the newly received subgradient $g_t$ into Alg. 1, which updates and maintains a low-rank sketch $\bar{G}_t$ of the AdaGrad preconditioning matrix $G_t$. We keep track of the cumulative escaped mass $\rho_{1:t}$, which we add back to the low-rank sketch to create the Sketchy preconditioner $\tilde{G}_t$, with which we perform the regular AdaGrad descent and projection.

---

**Algorithm 2** Sketchy AdaGrad (S-AdaGrad)

---

1: Input: constraint set $\mathcal{K}$, step size $\eta$, time horizon $T$.
2: Initialize $x_1 \in \mathcal{K}$, $\bar{G}_0 = \tilde{G}_0 = 0$.
3: **for** $t = 1, \dots, T$ **do**
4:     Play $x_t$, receive $g_t \in \partial f_t(x_t)$, suffer cost $f_t(x_t)$.
5:     Sketch $(\rho_t, \bar{G}_t) = \texttt{FD-update}(\bar{G}_{t-1}, g_t g_t^\top)$.
6:     Update $\tilde{G}_t = \bar{G}_t + \rho_{1:t} I$, $y_{t+1} = x_t - \eta \tilde{G}_t^{-1/2} g_t$, and $x_{t+1} = \underset{x \in \mathcal{K}}{\operatorname{argmin}} \|y_{t+1} - x\|_{\tilde{G}_t^{1/2}}^2$.
7: **end for**

---

**Theorem 3.** *Define* $\Omega_\ell = \min_{k < \ell} (\ell - k)^{-1} \sum_{i=k+1}^d \lambda_i(G_T)$, *then with* $\eta = \frac{D}{\sqrt{2}}$, *Alg. 2 guarantees the following additive regret bound:*

$$Regret_T(\texttt{S-AdaGrad}) \le D \left( \sqrt{2} \operatorname{tr} G_T^{1/2} + d\sqrt{\frac{\Omega_\ell}{2}} \right) ,$$

*where* $D$ *is the diameter of the constraint set* $\mathcal{K}$ *if* $\mathcal{K}$ *is bounded and* $\max_{t \in [T]} \|x_t - x^*\|_2$ *otherwise.*

*Proof.* See Sec. B.4.1. □

Notably in Thm. 3, $Regret_T = O\left(\sqrt{T}\right)$ and the lower eigenvalue dependence $\Omega_\ell$ is additive.

**Corollary 4.** *We can improve Theorem 3 slightly to*

$$Regret_T(\texttt{S-AdaGrad}) \le D \left( \sqrt{2} \operatorname{tr} G_T^{1/2} + \sqrt{\frac{d(d - \ell)\Omega_\ell}{2}} \right) .$$

*Proof.* See Sec. B.4.2. □

The regret bound above holds under the optimal tuning of the learning rate, which depends on problem quantities that can be unknown a priori. It is possible to design a parameter-free variant of Alg. 2 by using the norm $\|x\|_t = (x^\top (\tilde{G}_t + I)^{1/2} x)^{1/2}$ in the projection step of Alg. 2, as seen in [32].

## 4.2 FD for Shampoo

In this section, we adapt `FD-update` to Shampoo [5]. For simplicity, we optimize over $\mathbb{R}^{m \times n}$ in Alg. 3; projection may be handled as in Alg. 2. Similar to Sketchy AdaGrad, Sketchy Shampoo uses the FD approach outlined in Alg. 1 to sketch the left and right preconditioning matrices for Shampoo. In particular, we maintain two parallel sketching streams using Alg. 1 to produce sketches $\bar{L}_t, \bar{R}_t$ for the left and right preconditioning matrices. We keep track of the cumulative escaped masses $\rho_{1:t}^L$ and $\rho_{1:t}^R$ from sketching the left and right preconditioning matrices, respectively, and compensate the cumulative escaped mass to create the left and right Sketchy preconditioning matrices $\tilde{L}_t, \tilde{R}_t$.

---

**Algorithm 3** Sketchy Shampoo (S-Shampoo)

---

1: Input: step size $\eta$, time horizon $T$.
2: Initialize $X_0 = 0_{m \times n}$, $\tilde{L}_0 = \varepsilon I_m$, $\tilde{R}_0 = \varepsilon I_n$, $\bar{L}_0 = 0_m$, $\bar{R}_0 = 0_n$.
3: **for** $t = 1, \ldots, T$ **do**
4:    Play $X_t$, suffer $f_t(X_t)$, receive $G_t \in \partial f_t(X_t)$.
5:    Sketch $(\rho_t^L, \bar{L}_t) = $ `FD-update`$(\bar{L}_{t-1}, G_t G_t^\top)$, $(\rho_t^R, \bar{R}_t) = $ `FD-update`$(\bar{R}_{t-1}, G_t^\top G_t)$.
6:    Update $\tilde{L}_t = \bar{L}_t + \rho_{1:t}^L I_m$, $\tilde{R}_t = \bar{R}_t + \rho_{1:t}^R I_n$ and $X_{t+1} = X_t - \eta \tilde{L}_t^{-1/4} G_t \tilde{R}_t^{-1/4}$.
7: **end for**

---

Denote $L_T \stackrel{\text{def}}{=} \sum_{t=1}^T G_t G_t^\top + \varepsilon I$ and $R_T \stackrel{\text{def}}{=} \sum_{t=1}^T G_t^\top G_t + \varepsilon I$.

**Theorem 5.** *Suppose $G_1, \ldots G_T$ have rank at most $r$. Then Alg. 3 run with $\eta = D/\sqrt{2r}$ guarantees the following regret bound:*

$$Regret_T(\text{S-Shampoo}) \leq \sqrt{2r} D \left( \text{tr}(L_T^{1/4}) + m \Omega_{L,\ell}^{1/4} \right) \left( \text{tr}(R_T^{1/4}) + n \Omega_{R,\ell}^{1/4} \right) \ ,$$

*where $D = \max_{t \in [T]} \|X_t - X^*\|_F$ and $\Omega_{L,\ell}, \Omega_{R,\ell}$ are analogous bounds for $\rho_{1:T}^L, \rho_{1:T}^R$ from Lem. 1.*

*Proof.* See Sec. B.5.1. □

Bounds may be improved analogous to Cor. 4 for Alg. 3, but we omit the similar statement due to space.

## 4.3 Exponentially Weighted FD

This section discusses the modification of Alg. 1 to support exponential moving averages. Early in algorithm development, we noticed that attempting to approximate the unweighted sum of factored gradient covariances $\sum_t G_t G_t^\top$ and $\sum_t G_t^\top G_t$ with FD tended to an estimate of covariance that was roughly 0, creating numerical instabilities. Note that FD guarantee (Lem. 1) still holds—but the error term $\rho_{1:T}$ becomes greater than $\|G_T\|$, resulting in a vacuous bound due to lack of spectral decay.

Indeed, Fig. 3 motivating this work only confirmed that the exponential moving average $L_t(\beta_2) = \sum_t \beta_2^{T-t} G_t G_t^\top$ exhibits fast spectral decay (and analogously for $R_t$). Luckily, thanks to the recursion $L_{t+1}(\beta_2) = \beta_2 L_t + G_{t+1} G_{t+1}^\top$, the FD sketch may easily be adopted for this setting.

**Observation 6.** *Given a stream $g_t$ of vectors for $t \in [T]$, sketch size $\ell$, updates $(\rho_t^{(\beta_2)}, \bar{G}_t^{(\beta_2)}) = $ `FD-update`$(\beta_2 \bar{G}_{t-1}^{(\beta_2)}, g_t g_t^\top)$, and $G_T^{(\beta_2)} = \sum_{t=1}^T \beta_2^{T-t} g_t g_t^\top$, we have*

$$\left\| \bar{G}_T^{(\beta_2)} - G_T^{(\beta_2)} \right\| \leq \rho_{1:T}^{(\beta_2)} \leq \min_{k < \ell} \frac{\sum_{i=k+1}^d \lambda_i \left( G_T^{(\beta_2)} \right)}{\ell - k} \ .$$

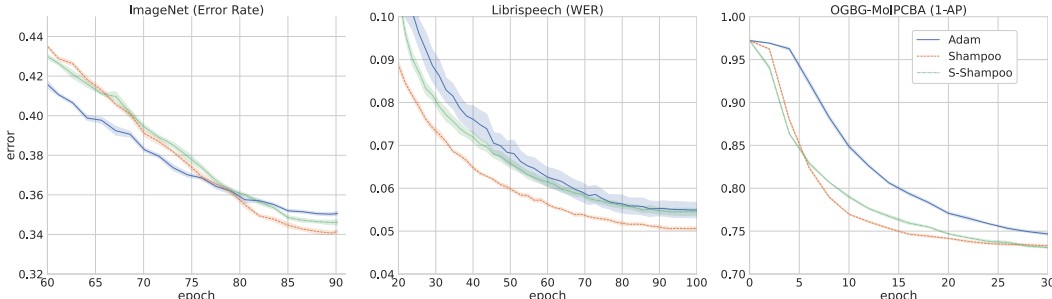

Figure 2: Test metrics are classification error rate for top-1 accuracy for ImageNet v2, word error rate for Librispeech, and one minus average precision for OGBG-MolPCBA. We plot the mean of 5 random seeds, with 1.96 times the standard error as error bars. For readers familiar with ImageNet v1, final validation accuracy for Shampoo was 77.69% (0.03%), S-Shampoo having 77.18% (0.04%), and Adam having 76.76% (0.03%), but we emphasize that due to tuning, the test set performance pictured above should be of primary concern.

## 5 Experiments

We investigate how much of Shampoo's quality our low-memory approach can recover (Sec. 5.1) and whether the factored covariance exhibits spectral decay amenable to sketching (Sec. 5.2). We relegate convex examples to Appendix A due to space constraints, which check that Sketchy compares favorably to related work in a setting directly related to the theory.

### 5.1 Deep Neural Networks

We evaluate the effectiveness of S-Shampoo as a practical second-order algorithm for training networks, including

- ResNet-50 [3] for ImageNet image classification task of ImageNet [33] with random cropping and flipping augmentations.
- A 16-layer Conformer model [34] for the audio transcription task, Librispeech [35].
- A GNN with 5 message-passing steps [36] on ogbg-molpcba [37], which classifies structural properties of graphically encoded molecule inputs.

Our FD variant of Shampoo introduces only one new hyperparameter, the rank $\ell$, which we do not tune, but set to $\ell = 256$, which translates to $4\times$ memory savings for Shampoo blocks of size $1024$ for the accumulators. Shampoo, Adam, and the underlying architectures introduce their own hyperparameters. S-Shampoo inherits those of Shampoo. We tune only common parameters between the three optimizers with the same budgets, selecting based on validation set accuracy. In the ImageNet case, we evaluate final test set performance using ImageNet v2 [38], as the ImageNet test set is unavailable. Additinoal training information is available in Appendix C.

As Fig. 2 demonstrates, the second-order information leveraged by Shampoo results in improvements over Adam, a first-order method. Our method performs at least as well as Adam in all cases, **despite using asympotically less memory to represent covariance** (as Fig. 1 shows, Adam uses $O(mn)$ for a rectangular weight matrix's diagonal accumulators, whereas S-Shampoo uses $O(mk + nk)$). In the GNN case (OBGB-MolPCBA), Shampoo does not perform as well; its training curves indicate overfitting, but we note that S-Shampoo was less susceptible to overfit like Adam.

### 5.2 Spectral Analysis

To explain Sketchy's strong performance in Sec. 5.1, we inspect the exponential moving average of Kronecker-factored gradient covariance for fast spectral decay. We find that this is indeed the case in practice, so Sketchy's low-rank plus diagonal covariance is representative of true training statistics.

For all our architectures, we tune Shampoo and extract the intermediate gradient covariances over the course of training. To make our curves comparable across architectures, we fix the parameter for

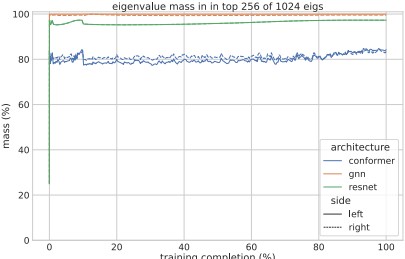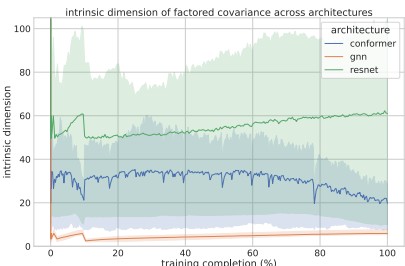

Figure 3: Two measures of spectral decay. As described in Sec. 5.2, we demonstrate spectral decay of $L_t$ and $R_t$ covariance factors of shape $1024 \times 1024$. On the left, we take the factors $C$ from the first layer of each network and plot the proportion of spectral mass captured by its top 256 eigenvalues throughout training, i.e., $\sum_{i=1}^{256} \lambda_i(C) / \sum_{i=1}^{1024} \lambda_i(C)$. On the right, we plot a continuous measure of eigenvalue concentration, the intrinsic dimension $\operatorname{tr} C / \lambda_{\max}(C)$, typically $10\times$ smaller than nominal dimension. We plot the average across all covariances for each network's weight's dimensions, with the shaded regions capturing the interquartile range.

the second moment, $\beta_2 = 0.999$ for these runs. Furthermore, ResNet-50 has a few parameters with dimension 2048, but the largest dimension for any parameter from the other two architectures is 1024, so we use the Blocked Shampoo variant discussed in Sec. 3.4 with block size 1024. In other words, weights containing a dimension 2048 are split into two. We tune other Shampoo parameters for each architecture, and plot statistics of Kronecker factors $L_t = \sum_i \beta_2^{t-i} G_t G_t^\top$ and $R_t = \sum_i \beta_2^{t-i} G_t^\top G_t$.

In Fig. 3, we plot the intrinsic dimension of Kronecker covariance factors over training for our three settings. The intrinsic dimension determines the rate at which empirical covariance estimates concentrate to their expectation, rather than a random vector's actual dimension, up to logarithmic factors (Vershynin [39], Remark 5.6.3). Despite actual dimensionality being over 1024, intrinsic dimension across all architectures stays below 105. A conspicuous phase shift 10% of the way through training may be the result of a change from linear learning rate warmup to a learning rate decay, starting at roughly 5% of the way into training.

Given $\beta_2 = 0.999$, we emphasize that the behavior in Fig. 3 is an emergent property of DL training. Though surely a lower $\beta_2$ would naturally result in lower intrinsic dimension (which can still be taken advantage of by Alg. 2 and 3), we would still expect higher intrinsic dimension if covariances were near-isometries. If we observe some large number $n = 10000$ draws $x_i$ of $1024 \times d$ matrices with iid $N(0, 1)$ entries, then numerical experiments show that the average intrinsic dimension of $\sum_{i=0}^{n-1} \beta_2^i x_i x_i^\top$ is 324.63 (0.52) and 862.13 (0.25) for $d = 1, 64$, respectively, with parenthesized numbers denoting standard error across 20 trials. Values generated this way are larger than the average intrinsic dimension of roughly 10, 30, 50 observed in Fig. 3.

## 6 Discussion

Up to spectral error, Alg. 2 achieves full-matrix AdaGrad regret despite approximating the *smallest* part of the spectrum of $G_t^{-1/2}$ at each step. Remarkably, these eigenvectors correspond to the *most* easily discernible signals of the covariance for the stream $g_t$. This apparent (and fortuitous) coincidence is resolved by considering the covariance of $\tilde{G}_t^{-1/2} g_t$: whitening the gradient to facilitate optimization best reflects on regret; as a result, approximating top eigenvectors of $G_T$ helps more than the bottom ones.

Our initial implementation focused on correctness rather than physical speed or memory reduction. Engineering optimizers competitive with existing industrial-strength implementations of Adam and Shampoo was out of scope. In implementing FD, we performed updates via the factored SVD of $[\beta_2^{1/2} B_t; G_t]$ rather than the eigendecomposition depicted in Alg. 1; this avoids squaring, which is unavoidable in Shampoo. For speed, Shampoo subsamples gradients for its covariance estimation and updates its inverse matrix roots intermittently, every fixed number of steps. A tuning script provided by Anil et al. [9] included gradients from every step, but updated roots every 10 steps. Since FD does

not separate sampling from its computation of estimated covariance eigendecomposition, we took the more difficult setting for S-Shampoo, only allowing it to simultaneously observe every $10^{\text{th}}$ gradient and update its covariance inverse roots (see Appendix G for a theoretical justification).

Though step-skipping makes Shampoo and S-Shampoo tractable, future work may explore further speedups: since FD only requires the top $\ell$ eigenvalues, iterative Lanczos-like routines which are accelerator-friendly, such as LOBPCG [40], may allow incremental updates to $\tilde{G}_t^{-1/2}$ in factored form with only a few matrix multiplies, S-Shampoo may be able to update more frequently than its non-sketched counterpart, further improving quality.

# 7 Conclusion

In this work, we address a gap in the OCO literature for low-memory optimization with the novel Alg. 2 and demonstrate its relevance to practical non-convex problems such as neural net training (Sec. 5.1) by leveraging a new observation about gradient covariance (Sec. 5.2).

The growing disparity between compute capability and memory bandwidth [11] underscores the need for further research in this direction. Further, large-batch settings reduce the performance gap between first and Shampoo-based second order methods, since the batch-size independent runtime of the optimizer is amortized per example used for the gradient calculation. Even in performing experiments for this work, we would frequently find that faster accelerators were unavailable, but many previous-generation ones were, encouraging us to leverage data-parallel training. For datasets such as Imagenet, we notice the advantage of second order methods in dealing with large batches even at relatively modest sizes, such as 1024; many works on explore several larger multiples of this [41].

Potential for future work includes numerical methods outlined in the previous section as well optimizing the rank $\ell$ across the many tensors in a network, as the spread in Fig. 3 highlights the large variance in covariance intrinsic dimension. Furthermore, the inductive biases conferred by the minima which different-rank representations of curvature reach may have problem-dependent generalization implications, a question which we leave for future work. For a comparison of full rank preconditioning's effect versus first-order minima, see Amari et al. [42].

# 8 Acknowledgments

Elad Hazan acknowledges funding from the Office of Naval Research grant N000142312156, the NSF award 2134040, and Open Philanthropy.

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
