Table 2: Dataset statistics for simple online convex examples. We stream through each example, providing a logistic loss over the binary label and linear predictor generated by an OCO learner. The feature count includes an all-constant intercept column as well. Datasets were retrieved from Chang and Lin [44].

| Dataset Name | Number of Examples | Number of Features |
|---|---|---|
| gisette_scale | 6000 | 5001 |
| a9a | 32561 | 124 |
| cifar10 | 50000 | 3073 |

Table 3: Average cumulative online loss across datasets and algorithms, ranked from lowest (1st place) to highest (6th place). Our proposal, S-Adagrad, is in bold.

| | Place | 1 | 2 | 3 | 4 | 5 | 6 |
|---|---|---|---|---|---|---|---|
| cifar10 | Alg. | RFD-SON | **S-Adagrad** | Adagrad | OGD | Ada-FD | FD-SON |
| | Loss | 0.297 | 0.297 | 0.303 | 0.308 | 2.999 | $6 \times 10^6$ |
| gisette | Alg. | **S-Adagrad** | RFD-SON | Ada-FD | Adagrad | OGD | FD-SON |
| | Loss | 0.158 | 0.167 | 0.196 | 0.209 | 0.224 | 2.432 |
| a9a | Alg. | Adagrad | **S-Adagrad** | OGD | RFD-SON | Ada-FD | FD-SON |
| | Loss | 0.332 | 0.333 | 0.335 | 0.335 | 0.354 | 0.539 |

# A    Online Convex Examples

In this section, we evaluate the performance of S-Adagrad in the classical online convex optimization setting.

We consider three different datasets, summarized in Tbl. 2. For each dataset, we evaluate several works based on the frequent directions sketch, Ada-FD [26], FD-SON [27], RFD-SON [43]. As baselines we also add online gradient descent (OGD) and diagonal Adagrad [2]. For each dataset, we consider the online convex loss of a logistic binary classification loss over a linear learner on the features in each dataset augmented with a constant feature for the intercept.

For methods which require a nonzero diagonal initial regularizer $\delta I$, namely FD-SON and Ada-FD, we tune $\delta$ in $10^{-6}, 10^{-5}, \cdots 1$ and the learning rate $\eta$ over the same range as well, for 49 hyperparameters total. For methods which have $\delta = 0$ (Adagrad, OGD, S-Adagrad, RFD-SON), we instead tune $\eta$ on 49 points spaced evenly on the same logarithmic scale, $[10^{-6}, 1]$ to fairly allocate hyperparameter training budget. Note that the variant of RFD-SON, $\text{RFD}_0$, which sets $\delta = 0$ is the main variant evaluated by Luo et al. [43]. The sketch size was fixed to be 10 throughout.

We sort and display cumulative average regret in Tbl. 3 and Fig. 4. S-Adagrad is the only method to consistently place among the top three across all datasets. We suspect that a combination of allowing $\delta = 0$ (removing inductive bias about regularization; notice in Tbl. 3 that Ada-FD and FD-SON, the two methods with $\delta > 0$, routinely place last) and the ability to deal with the effectively zero exp-concavity constant of the logistic loss [45] explain S-Adagrad's performance.

---

[1]The regret of Ada-FD is expressed in terms of dynamic run-time quantities which do not admit a universal bound in terms of $G_T$; we display its regret for the specific case of Observation 2 instead (a detailed look at its regret is given in Appendix B.3).

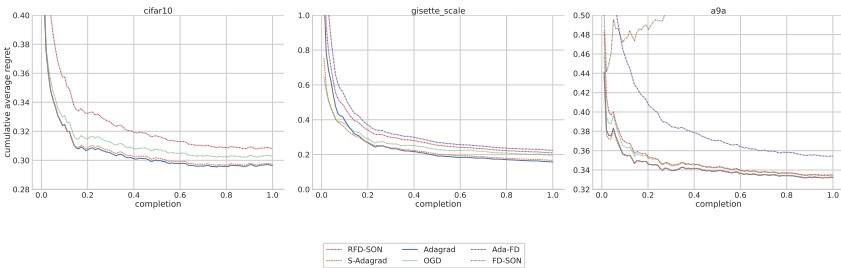

Figure 4: Cumulative average regret curves for logistic loss as a function of percent of dataset completion in a single online pass, resummarizing Tbl. 3 visually.

# B   Proof details

**Notation.**   Let $\|\cdot\|$ denote the $\ell_2$ norm for a vector. Let $\|\cdot\|_{op}$, $\|\cdot\|_F$ denote the operator norm and the Frobenius norm of a matrix, respectively. For a positive definite matrix $A$, we use $\|x\|_A = \sqrt{x^\top A x}$ to denote the matrix norm induced by $A$, and $\|x\|_A^* = \sqrt{x^\top A^{-1} x}$ to denote the dual norm of the induced matrix norm. For a matrix $A$, $A^{-1}$ is the inverse of $A$ if $A$ is full rank; otherwise, $A^{-1}$ is taken to be the Moore-Penrose pseudoinverse. Finally, $\overline{\text{vec}}\,(\cdot)$ denotes the row-major vectorization of a given matrix, and $\otimes$ denotes the Kronecker product between two matrices

## B.1   Reduction from Non-convex Optimization to Online Convex Optimization

In this section, we give more details for the reduction of non-convex optimization to online convex optimization for completeness. We use the framework of [6], though many related results exist in the optimization literature. The algorithm is stated in Algorithm 4, where we optimize the non-convex function $f$ by iteratively optimizing subproblems $f_t$ that are strongly convex. Given any OCO algorithm $\mathcal{A}$, in each episode, we first construct $f_t$ and then use $\mathcal{A}$ to optimize it for $N$ time steps. We state the convergence guarantees in terms of the adaptive ratio, defined below.

---

**Algorithm 4** Reduction from Non-convex optimization to Online Convex Optimization

---

1: Input: initial $x_1$, time horizon $T$, episode length $N$, smoothness parameter $L$, online convex optimization algorithm $\mathcal{A}$.
2: **for** $t = 1, \ldots, T$ **do**
3:     Construct $f_t(x) = f(x) + L\|x - x_t\|^2$.
4:     Initialize $\mathcal{A}$ to start at $x_t$, and set $x_t^1 = x_t$.
5:     **for** $n = 1, \ldots, N$ **do**
6:         Play $x_t^n$, receive stochastic gradient $\tilde{\nabla} f_t(x_t^n)$, construct $g_t^n(x) = \tilde{\nabla} f_t(x_t^n)^\top(x)$.
7:         Update $x_t^{n+1} = \mathcal{A}(g_t^1, \ldots, g_t^n)$.
8:     **end for**
9:     Update $x_{t+1} = \frac{1}{N}\sum_{n=1}^{N} x_t^n$.
10: **end for**
11: Output iterate $x_{t^*} = \operatorname{argmin}_{t \in [T+1]} \|\nabla f(x_t)\|$.

---

**Definition 7.** (Adaptive ratio.) Let $\mathcal{A}$ be an algorithm, and consider a convex function $f$. Given a stochastic gradient oracle with variance bounded by $\sigma^2$, let $x_{\mathcal{A}}$ be the output of $\mathcal{A}$ with at most $T$ oracle calls, and let $x^* \in \operatorname{argmin}_x f(x)$. Define the adaptive ratio of $\mathcal{A}$ as

$$\mu_{\mathcal{A}}(f) = \frac{f(x_{\mathcal{A}}) - f(x^*)}{\|x_1 - x^*\|\frac{\sigma}{\sqrt{T}}}.$$

The adaptive ratio captures the performance of $\mathcal{A}$ relative to SGD. For certain algorithms, such as AdaGrad [2], the adaptive ratio can be as small as $\frac{1}{\sqrt{d}}$. For more discussions on this notion see [6].

**Theorem 8.** *(Theorem A.2 in [6]) Consider a non-convex function $f$, and suppose $f$ is $L$-smooth and bounded: $\|\nabla^2 f(x)\|_2 \leq L$ and $\max_{x,y} f(x) - f(y) \leq M$. Additionally suppose we have access to a stochastic gradient oracle with variance bounded by $\sigma^2$. Let $\mu = \max_t \mu_{\mathcal{A}}(f_t)$. Then Algorithm 4 run with $T = \frac{12ML}{\epsilon^2}$ and $N = \frac{48\mu^2\sigma^2}{\epsilon^2}$ returns a point $x_{t^*}$ such that*

$$\mathbb{E}[\|\nabla f(x_{t^*})\|] \leq \varepsilon.$$

*The total number of calls to the stochastic gradient oracle is bounded by $T \cdot N = O(\mu^2\sigma^2/\epsilon^4)$.*

## B.2   Proof of Lem. 1

*Proof.* Let $H_T \in \mathbb{R}^{T \times d}$ denote the matrix of stacked gradients, where the $t$-th row of $H_T$ is $g_t$. Then $H_T^\top H_T = G_T$, and FD iteratively sketches $H_T$. Let $H_T = U\Sigma V^\top$ be the SVD of $H_T$, and let $H_{T,k} = U_k \Sigma_k V_k^\top$ denote the best rank-$k$ approximation of $H_T$, where $U_k, V_k$ are the first $k$ columns

of the matrices, and $\Sigma_k$ is the upper left $k \times k$ submatrix of $\Sigma$. By the proof of Theorem 1.1 in [13], we have

$$\rho_{1:T} \leq \min_{k < \ell} \frac{\|H_T - H_{T,k}\|_F^2}{\ell - k} = \min_{k < \ell} \frac{\sum_{i=k+1}^d \lambda_i(H_T^\top H_T)}{\ell - k} = \min_{k < \ell} \frac{\sum_{i=k+1}^d \lambda_i(G_T)}{\ell - k} \leq \sum_{i=\ell}^d \lambda_i(G_T) ,$$

where the last inequality follows by choosing $k = \ell - 1$. $\qquad\square$

### B.3 Proof of Observation 2

*Proof.* Let $\Sigma = \mathbb{E}\left[g_t g_t^\top\right]$ denote the covariance of the gradients, and $\lambda_i = \lambda_i(\Sigma)$ denote its $i$-th eigenvalue. By definition, $g_t$ has the following distribution: $g_t = w_i$ with probability $\lambda_i$. At iteration $t$, we have the current sketch $\bar{G}_{t-1} \in \mathbb{R}^{\ell \times d}$, and we receive the new gradient $g_t$. Ada-FD uses $\bar{G}_{t-1} + \delta I$ as their preconditioner.

We first show that under the distribution of the cost functions, if $\bar{G}_{t-1}$ has rank $\ell - 1$, then $\mathbb{E}\left[\rho_t | \bar{G}_{t-1}\right] \geq \sum_{i=\ell}^r \lambda_i$. Let $\bar{G}_{t-1} = U\Sigma V^\top$ be the SVD of $\bar{G}_{t-1}$, and $v_i$ be the $i$-th row of $V \in \mathbb{R}^{\ell-1 \times d}$. Let $N_{t-1} = W \setminus \{v_1, \ldots, v_{\ell-1}\}$ be the set of basis vectors not in the row space of $\bar{G}_{t-1}$, then $|N_{t-1}| = r - \ell + 1$. If $g_t \in \mathrm{span}(v_1, \ldots, v_\ell)$, then $\rho_t = 0$; otherwise $\rho_t = 1$, with probability $\sum_{i:w_i \in N_{t-1}} \lambda_i \geq \sum_{i=\ell}^r \lambda_i$.

We proceed to bound the probability that $\bar{G}_{t-1}$ has rank $\ell' \leq \ell - 2$. Note that this event is equivalent to having fewer than $\ell - 1$ distinct vectors drawn from $W$. Let $I_i$ be the indicator variable for drawing $w_i$ in the first $t - 1$ rounds, then we can obtain the expected number of distinct vectors as follows

$$\mathbb{E}\left[\sum_{i=1}^r I_i\right] = \sum_{i=1}^r \mathbb{E}\left[I_i\right] = \sum_{i=1}^r 1 - (1 - \lambda_i)^{t-1}.$$

We consider the random variable $r - \sum_{i=1}^r I_i$, and by Markov's inequality,

$$\mathbb{P}\left[r - \sum_{i=1}^r I_i \geq r - \ell + 2\right] \leq \frac{\sum_{i=1}^r (1 - \lambda_i)^{t-1}}{r - \ell + 2} \leq \frac{r(1 - \lambda_r)^{t-1}}{2} .$$

Note that this is exactly the probability of having fewer than $\ell - 1$ distinct vectors in the first $t - 1$ draws. We conclude that for $t \geq \log r / \lambda_r + 1$, $\mathbb{P}\left[\mathrm{rank}(\bar{G}_{t-1}) = \ell - 1\right] \geq \frac{1}{2}$. This implies that $\mathbb{E}\left[\rho_t\right] \geq \sum_{i=\ell}^r \lambda_i / 2$ after an initial log number of rounds, and assuming $T \geq 2 \log r / \lambda_r$,

$$\mathbb{E}\left[\sum_{t=1}^T \rho_t\right] \geq \mathbb{E}\left[\sum_{t=T/2+1}^T \rho_t\right] \geq \frac{T}{4} \sum_{i=\ell}^r \lambda_i.$$

Similarly,

$$\mathbb{E}\left[\sum_{t=1}^T \sqrt{\rho_t}\right] \geq \mathbb{E}\left[\sum_{t=T/2+1}^T \sqrt{\rho_t}\right] \geq \frac{T}{4} \sum_{i=\ell}^r \lambda_i,$$

where the second inequality holds because $\rho_t = 0$ or 1, so $\sqrt{\rho_t} = \rho_t$ for all $t$. The quantities $\sum_{t=1}^T \rho_t$ and $\sum_{t=1}^T \sqrt{\rho_t}$ correspond to $\Delta_T$ and $\sum_{t=1}^T \sqrt{\sigma_t}$ in Theorem 1 of [26], respectively. Therefore, under our setting, the expectation of the upper bound in Theorem 1 is at least

$$\eta \, \mathbb{E}\left[\max\left\{1, \frac{1 + \sqrt{\sum_{t=1}^T \rho_t}}{\delta}\right\} \mathrm{tr}(G_T^{1/2})\right] + \frac{D^2}{2\eta} \mathbb{E}\left[\sum_{t=1}^T \sqrt{\rho_t}\right]. \qquad (1)$$

If we can tune $\delta$, then the max function evaluates to at least 1, and

$$(1) \geq \eta \, \mathbb{E}\left[\mathrm{tr}(G_T^{1/2})\right] + \frac{D^2}{2\eta} \frac{T}{4} \sum_{i=\ell}^r \lambda_i \geq \eta \, \mathbb{E}\left[\sqrt{\mathrm{tr}(G_T)}\right] + \frac{D^2 T}{8\eta} \sum_{i=\ell}^r \lambda_i = \eta\sqrt{T} + \frac{D^2 T}{8\eta} \sum_{i=\ell}^r \lambda_i,$$

where the last equality holds because $\mathrm{tr}(G_T) = \sum_{t=1}^T \|g_t\|_2^2 = T$. Optimizing $\eta$, we conclude that the regret upper bound for Ada-FD is $\Omega(T^{3/4})$ in expectation.

$\qquad\square$

### B.4 Proof details for Section 4.1, S-AdaGrad

#### B.4.1 Proof of Theorem 3

The following observations are made of Algorithm 1:

**Observation 9.** *Denote by* $\bar{U}_t \overset{def}{=} \left[ U_t; U_t^{\perp} \right]$. *If each $M_t$ is of rank 1, then*

$$\bar{G}_t + \lambda_{\ell}^{(t)} I = \bar{G}_{t-1} + M_t + \lambda_{\ell}^{(t)} N_t \ ,$$

*where* $N_t = U_t^{\perp} \left( U_t^{\perp} \right)^{\top}$.

*Proof.* By definition of Algorithm 1, $\bar{G}_{t-1} = B_{t-1} B_{t-1}^{\top}$ is of rank at most $\ell - 1$. Under the assumption that $M_t$ is of rank 1, $\bar{G}_{t-1} + M_t$ is of rank at most $\ell$. Therefore, $\lambda_{\ell+1:d} = 0$. Then, we have the following:

$$
\begin{aligned}
\bar{G}_t + \lambda_{\ell}^{(t)} I &= U_t \operatorname{diag} \left( \lambda_{[1:\ell]}^{(t)} - \lambda_{\ell}^{(t)} \right) U_t^{\top} + \lambda_{\ell}^{(t)} I \\
&= U_t \operatorname{diag} \left( \lambda_{[1:\ell]}^{(t)} - \lambda_{\ell}^{(t)} \right) U_t^{\top} + \lambda_{\ell}^{(t)} (U_t U_t^{\top} + N_t) \\
&= U_t \operatorname{diag} \lambda_{[1:\ell]}^{(t)} U_t^{\top} + \lambda_{\ell}^{(t)} N_t \\
&= \bar{U}_t \operatorname{diag} \lambda^{(t)} \bar{U}_t^{\top} + \lambda_{\ell}^{(t)} N_t \\
&= \bar{G}_{t-1} + M_t + \lambda_{\ell}^{(t)} N_t \ .
\end{aligned}
$$

$\square$

**Lemma 10.** *Let $\lambda_{\ell}^{(s:t)}$ denote $\sum_{j=s}^{t} \lambda_{\ell}^{(j)}$. Let $\tilde{G}_t \overset{def}{=} \bar{G}_t + \lambda_{\ell}^{(1:t)} I$, $G_t = \sum_{s=1}^{t} M_s$, where each $M_t$ is of rank 1. Let the initial $G_0 = \bar{G}_0 = 0$, then the following relation between $\tilde{G}_t$ and $G_t$ holds for all t:*

$$\tilde{G}_t = G_t + \sum_{s=1}^{t} \lambda_{\ell}^{(s)} N_s \ .$$

*Proof.* The lemma follows from induction on $t$. Base case $\tilde{G}_0 = G_0$ holds by definition. Suppose the above equation holds for $t - 1$. Then,

$$
\begin{aligned}
\tilde{G}_t = \bar{G}_t + \lambda_{\ell}^{(1:t)} I &=_1 \bar{G}_{t-1} + M_t + \lambda_{\ell}^{(t)} N_t + \lambda_{\ell}^{(1:t-1)} I \\
&= \tilde{G}_{t-1} + M_t + \lambda_{\ell}^{(t)} N_t \\
&=_2 G_{t-1} + \sum_{s=1}^{t-1} \lambda_{\ell}^{(s)} N_s + M_t + \lambda_{\ell}^{(t)} N_t \\
&= G_t + \sum_{s=1}^{t} \lambda_{\ell}^{(s)} N_s \ ,
\end{aligned}
$$

where $=_1$ follows from Observation 9 and $=_2$ follows from induction hypothesis. $\square$

**Remark 11.** Note that the above lemma immediately provides an approximate isometry $\bar{G}_t \preceq G_t \preceq \tilde{G}_t$.

Now, we return to the proof of Thm. 3.

*Proof.* First, we make the following observation of Algorithm 2:

**Observation 12.** *By specification of Algorithm 1,2, $g_t \in span(\tilde{G}_t)$, $\forall t$.*

We follow the standard AdaGrad [2, 15] analysis. By algorithm specification,

$$y_{t+1} - x^* = x_t - x^* - \eta \tilde{G}_t^{-1/2} g_t ,$$

$$\tilde{G}_t^{1/2}(y_{t+1} - x^*) = \tilde{G}_t^{1/2}(x_t - x^*) - \eta \tilde{G}_t^{1/2} \tilde{G}_t^{-1/2} g_t =_1 \tilde{G}_t^{1/2}(x_t - x^*) - \eta g_t ,$$

where $=_1$ follows from Observation 12.

With standard AdaGrad analysis, we can bound regret $\text{Regret}_T$ above by the sum of the diameter bound and the gradient bound:

$$\underbrace{\frac{1}{2\eta} \sum_{t=1}^{T} \|x_t - x_*\|^2_{\tilde{G}_t^{1/2} - \tilde{G}_{t-1}^{1/2}}}_{R_D} + \underbrace{\frac{\eta}{2} \sum_{t=1}^{T} \|g_t\|_{F_t^{-1/2}}}_{R_G} .$$

Note that by algorithm specification, we have $\forall t$,

$$\tilde{G}_t = \bar{G}_t + \rho_{1:t} I =_1 \tilde{G}_{t-1} + g_t g_t^\top + \rho_t N_t \succeq \tilde{G}_{t-1} + g_t g_t^\top ,$$

where $=_1$ follows from the proof of Lemma 10. In particular, $\tilde{G}_t \succeq \tilde{G}_{t-1}$.

Using Remark 11, the gradient norm term in the regret bound can be further bounded by

$$R_G = \frac{\eta}{2} \sum_{t=1}^{T} g_t^\top \tilde{G}_t^{-1/2} g_t \leq \frac{\eta}{2} \sum_{t=1}^{T} g_t^\top G_t^{-1/2} g_t \leq \eta \operatorname{tr}\left(G_T^{1/2}\right) ,$$

where the last inequality follows from Lemma 10 of Duchi et al. [2].[2] The diameter norm term in the regret bound can be bounded by

$$R_D = \frac{1}{2\eta} \sum_{t=1}^{T} \|x_t - x_*\|^2_{\tilde{G}_t^{1/2} - \tilde{G}_{t-1}^{1/2}} \leq_1 \frac{D^2}{2\eta} \operatorname{tr}\left(\tilde{G}_T^{1/2}\right)$$

$$\leq \frac{D^2}{2\eta} \operatorname{tr}\left((G_T + \rho_{1:T} I)^{1/2}\right)$$

$$\leq_2 \frac{D^2}{2\eta} \left(\operatorname{tr} G_T^{1/2} + \operatorname{tr}(\rho_{1:T} I)^{1/2}\right) ,$$

where $\leq_1$ follows from monotonicity of $\tilde{G}_t$'s, $\|\cdot\|_{op} \leq \operatorname{tr}(\cdot)$ for positive semidefinite matrices, and linearity of $\operatorname{tr}(\cdot)$, and $\leq_2$ follows from that for $X \in \mathbb{R}^d$, $X \succeq 0$, $\operatorname{tr}(X + \sigma I_d)^{1/2} \leq \operatorname{tr}(X^{1/2}) + d\sqrt{\sigma}$. Combining, we have

$$\text{Regret}_T \leq \frac{d\sqrt{\rho_{1:T}} + \operatorname{tr} G_T^{1/2}}{2\eta} D^2 + \eta \operatorname{tr}\left(G_T^{1/2}\right) = D\left(\sqrt{2} \operatorname{tr} G_T^{1/2} + d\sqrt{\frac{\rho_{1:T}}{2}}\right) ,$$

where the last equality is established by choosing $\eta = \frac{D}{\sqrt{2}}$. $\qquad\square$

### B.4.2 Proof of Corollary 4

*Proof.* Following the proof of Theorem 3, we have

$$\text{Regret}_T \leq \frac{D^2 \operatorname{tr} \tilde{G}_t^{1/2}}{2\eta} + \eta \operatorname{tr} G_T^{1/2} ,$$

Denote the accumulated error term

$$E = \sum_{t=1}^{T} \rho_t N_t .$$

---

[2]The FTL-BTL lemma [46] alone is not sufficient to justify this inequality, at least interpreting $G^{-1/2}$ as $\left(G^{1/2}\right)^+$. However, Duchi et al. [2] rely on concavity of $X \mapsto \operatorname{tr} X^{1/2}$ to show a semidefinite version of the statement.

Then, by Lemma 10 and sub-additivity of $\operatorname{tr}\left((\cdot)^{1/2}\right)$ [47],

$$\operatorname{Regret}_T \le \left(\frac{D^2}{2\eta} + \eta\right)\operatorname{tr} G_T^{1/2} + \frac{D^2}{2\eta}\operatorname{tr} E^{1/2},$$

where it remains to bound the last term. Let $Q$ be a matrix with column vectors $q_i$ that forms an eigenbasis of $E^{1/2}$; this diagonalizes $E$ as well. Notice that

$$\lambda_i(E) = \sum_{t=1}^T \rho_t q_i^\top N_t q_i \ ,$$

and since

$$\lambda_i^{1/2}(E) = \lambda_i\left(E^{1/2}\right) \ ,$$

that we can characterize

$$\operatorname{tr} E^{1/2} = \sum_{i=1}^d \lambda_i^{1/2}(E) = \sum_{i=1}^d \left(\sum_{t=1}^T \rho_t q_i^\top N_t q_i\right)^{1/2} \ .$$

Denote $u_{t,i} = q_i^\top N_t q_i$, since $N_t$ is a rank-$(d-\ell)$ projection, $\|u_t\|_1 = d - \ell$. Then $\operatorname{tr} E^{1/2}$ is upper bounded by the value of the program

$$\max_{u_{t,i}} \quad \sum_{i=1}^d \left(\sum_{t=1}^T \rho_t u_{t,i}\right)^{1/2}$$

$$\text{s.t.} \quad \sum_{i=1}^d u_{t,i} = d - \ell \qquad \forall t \in [T] \ .$$

Note that

$$\sum_{i=1}^d \left(\sum_{t=1}^T \rho_t u_{t,i}\right)^{1/2} \le_1 \sqrt{d}\sqrt{\sum_{i=1}^d \sum_{t=1}^T \rho_t u_{t,i}} = \sqrt{d}\sqrt{\sum_{t=1}^T \rho_t \sum_{i=1}^d u_{t,i}} =_2 \sqrt{d\rho_{1:T}(d-\ell)} \ ,$$

where $\le_1$ follows from Cauchy-Schwarz, and $=_2$ follows from the constraint on $\sum_{i=1}^d u_{t,i}$.

Combining, we have

$$\operatorname{Regret}_T \le \frac{\sqrt{d(d-\ell)\rho_{1:T}} + \operatorname{tr} G_T^{1/2}}{2\eta} D^2 + \eta \operatorname{tr} G_T^{1/2} = D\left(\sqrt{2}\operatorname{tr} G_T^{1/2} + \sqrt{\frac{d(d-\ell)\rho_{1:T}}{2}}\right) \ ,$$

where the last equality follows from the choice of step size $\eta = \frac{D}{\sqrt{2}}$.

$\square$

### B.5 Proof details for Section 4.2, S-Shampoo

#### B.5.1 Proof of Theorem 5

*Proof.* First, we establish the following observation and lemma analogous to Observation 9 and Lemma 10:

**Observation 13** (Analogous to Observation 9). *Let $V_t \Sigma_t^L V_t^\top = \bar{L}_{t-1} + G_t G_t^\top$ be the eigen-decomposition of the un-deflated sketch, where $V_t \in \mathbb{R}^{m\times m}$. Suppose $\operatorname{rank}(\Sigma_t^L) = k$, where $k \in [\ell-1, \ell-1+r]$. Write $V_t = [V_t^{\|} \ V_t^\perp]$, where $V_t^{\|}$ contain the first $k$ columns of $V_t$. Then by definition*

$$\bar{L}_t + \rho_t^L I \succeq \bar{L}_{t-1} + G_t G_t^\top + \rho_t^L V_t^\perp \left(V_t^\perp\right)^\top \ .$$

*Analogously for the right conditioner, let $W_t \Sigma_t^R W_t^\top = \bar{R}_{t-1} + G_t^\top G_t$, and write $W_t = [W_t^{\|} \ W_t^\perp]$, then*

$$\bar{R}_t + \rho_t^R I \succeq \bar{R}_{t-1} + G_t^\top G_t + \rho_t^R W_t^\perp \left(W_t^\perp\right)^\top \ .$$

**Lemma 14.** *(Analogous to Lemma 10) Define* $N_t^L = V_t^\perp \left(V_t^\perp\right)^\top, N_t^R = W_t^\perp \left(W_t^\perp\right)^\top$, *then*

$$\tilde{L}_t \succeq \sum_{s=1}^t G_s G_s^\top + \sum_{s=1}^t \rho_s^L N_s^L + \varepsilon I_m, \quad \tilde{R}_t \succeq \sum_{s=1}^t G_s^\top G_s + \sum_{s=1}^t \rho_s^R N_s^R + \varepsilon I_n.$$

We follow the shampoo proof in [5]. Let $x_t = \overline{\text{vec}}(X_t)$, $g_t = \overline{\text{vec}}(G_t)$, where $\overline{\text{vec}}(\cdot)$ denote the row-major vectorization of a given matrix.

Kronecker product $\otimes$ obeys the following properties as shown in [5]:

**Lemma 15** (Lemma 3,4 in Gupta et al. [5]). *For matrices $A, A', B, B'$ of appropriate dimensions and vectors $u, v, L \in \mathbb{R}^{m \times m}, R \in \mathbb{R}^{n \times n}, G \in \mathbb{R}^{m \times n}$, the following properties hold:*

1. $(A \otimes B)(A' \otimes B') = (AA') \otimes (BB')$.

2. $(A \otimes B)^\top = A^\top \otimes B^\top$.

3. $A, B \succeq 0, (A \otimes B)^{-1} = A^{-1} \otimes B^{-1}$.

4. $A \succeq A', B \succeq B'$, then $A \otimes B \succeq A' \otimes B'$.

5. $\text{tr}(A \otimes B) = \text{tr}(A) + \text{tr}(B)$.

6. $\overline{\text{vec}}(uv^\top) = u \otimes v$.

7. $(L \otimes R^\top)\overline{\text{vec}}(G) = \overline{\text{vec}}(LGR)$.

Then the shampoo update is

$$x_{t+1} = x_t - \eta(\tilde{L}_t^{1/4} \otimes \tilde{R}_t^{1/4})^{-1} g_t.$$

Let $\tilde{H}_t \overset{\text{def}}{=} \tilde{L}_t^{1/4} \otimes \tilde{R}_t^{1/4}$, then by Lemma 15, $\tilde{H}_t$ is monotone increasing with $t$, since $\tilde{L}_t$ and $\tilde{R}_t$ are monotone by Observation 13. Thus, by standard analysis [15] for Online Mirror Descent (OMD), we can break down the regret into the diameter bound and the gradient bound:

$$\text{Regret}_T \leq R_D + R_G, \quad \text{where}$$

f

$$R_D = \frac{1}{2\eta} \sum_{t=1}^T \left(\|x_t - x^*\|_{\tilde{H}_t}^2 - \|x_{t+1} - x^*\|_{\tilde{H}_t}^2\right), \quad R_G = \frac{\eta}{2} \sum_{t=1}^T \left(\|g_t\|_{\tilde{H}_t}^*\right)^2.$$

We proceed to bound $R_D$ and $R_G$ separately. For $R_D$,

$$\begin{aligned} R_D &\leq \frac{1}{2\eta} \sum_{t=1}^T \|x_t - x^*\|_{\tilde{H}_t - \tilde{H}_{t-1}}^2 + \|x_1 - x^*\|_{\tilde{H}_0}^2 \\ &\leq \frac{1}{2\eta} \sum_{t=1}^T \|\tilde{H}_t - \tilde{H}_{t-1}\|_{op} \|x_t - x^*\|_2^2 + \|x_1 - x^*\|_{\tilde{H}_0}^2 \\ &\leq_1 \frac{D^2}{2\eta} \sum_{t=1}^T \text{tr}(\tilde{H}_t - \tilde{H}_{t-1}) + \|x_1 - x^*\|_{\tilde{H}_0}^2 \\ &\leq \frac{D^2}{2\eta} \text{tr}(\tilde{H}_T), \end{aligned}$$

where $\leq_1$ holds since $\tilde{H}_t$'s are increasing in $t$, and we have for positive semidefinite matrices $\text{tr}(\cdot) \geq \|\cdot\|_{op}$.

Now we try to bound $R_G$. First, we have that

**Lemma 16** (Lemma 8 in Gupta et al. [5]). *If $G \in \mathbb{R}^{m \times n}$ with rank at most $r$, and $g = \overline{vec}(G)$, then $\forall \varepsilon \geq 0, \forall t$,*

$$\varepsilon I_{mn} + \frac{1}{r} \sum_{s=1}^{t} g_s g_s^\top \preceq \left( \varepsilon I_m + \sum_{s=1}^{t} G_s G_s^\top \right)^{1/2} \otimes \left( \varepsilon I_n + \sum_{s=1}^{t} G_s^\top G_s \right)^{1/2} .$$

Define $M_t^L \in \mathbb{R}^{m \times m}, M_t^R \in \mathbb{R}^{n \times n}$ by

$$M_t^L \stackrel{\text{def}}{=} \sum_{s=1}^{t} G_s G_s^\top + \sum_{s=1}^{t} \rho_s^L N_s^L + \varepsilon I_m , \quad M_t^R \stackrel{\text{def}}{=} \sum_{s=1}^{t} G_s^\top G_s + \sum_{s=1}^{t} \rho_s^R N_s^R + \varepsilon I_n ,$$

then by Lemma 14,

$$\tilde{L}_t \succeq M_t^L , \quad \tilde{R}_t \succeq M_t^R .$$

Observe that in addition,

$$M_t^L \succeq \varepsilon I_m + \sum_{s=1}^{t} G_s G_s^\top , \quad M_t^R \succeq \varepsilon I_n + \sum_{s=1}^{t} G_s^\top G_s .$$

Again by Lemma 15,

$$I_m \otimes \left( \varepsilon I_n + \sum_{s=1}^{t} G_s^\top G_s \right) \preceq I_m \otimes M_t^R , \quad \left( \varepsilon I_m + \sum_{s=1}^{t} G_s G_s^\top \right) \otimes I_n \preceq M_t^L \otimes I_n .$$

Combining, we have

$$\varepsilon I_{mn} + \frac{1}{r} \sum_{s=1}^{t} g_s g_s^\top \preceq \left( M_t^L \right)^{1/2} \otimes \left( M_t^R \right)^{1/2} \preceq \tilde{L}_t^{1/2} \otimes \tilde{R}_t^{1/2} .$$

Define $\hat{H}_t \succ 0 \ \forall t \in [T]$ by

$$\hat{H}_t \stackrel{\text{def}}{=} \left( r \varepsilon I_{mn} + \sum_{s=1}^{t} g_s g_s^\top \right)^{1/2} \preceq \sqrt{r} \tilde{H}_t .$$

The bound on $R_G$ depends on the following lemma:

**Lemma 17** (Lemma 2 in Gupta et al. [5]). *Consider a sequence of vectors $\{g_t\}_{t=1}^{T}$. Given a function $\Phi(\cdot)$ over positive semidefinite matrices,*

$$\sum_{t=1}^{T} \left( \|g_t\|_{H_t}^* \right)^2 \leq \sum_{t=1}^{T} \left( \|g_t\|_{H_T}^* \right)^2 + \Phi(H_T) - \Phi(H_0) ,$$

*where*

$$H_t = \underset{H \succ 0}{\operatorname{argmin}} \left\{ \left( \sum_{s=1}^{t} g_s g_s^\top \right) \cdot H^{-1} + \Phi(H) \right\} .$$

Let $\Phi(H) \stackrel{\text{def}}{=} \operatorname{tr}(H) + r \varepsilon \operatorname{tr}(H^{-1})$ and since

$$\underset{H \succ 0}{\operatorname{argmin}} \left\{ \left( \sum_{s=1}^{t} g_s g_s^\top \right) \cdot H^{-1} + \Phi(H) \right\} = \underset{H \succ 0}{\operatorname{argmin}} \left\{ \operatorname{tr}\left( \hat{H}_t^2 H^{-1} + H \right) \right\} = \hat{H}_t ,$$

the above lemma gives

$$\sum_{t=1}^{T} \left( \|g_t\|_{\hat{H}_t}^* \right)^2 \leq \sum_{t=1}^{T} \left( \|g_t\|_{\hat{H}_T}^* \right)^2 + \Phi(\hat{H}_T) - \Phi(\hat{H}_0) \leq 2 \operatorname{tr}(\hat{H}_T) ,$$

which by inequality of $\hat{H}_t$ and $\tilde{H}_t$ established above, gives

$$R_G \overset{\text{def}}{=} \frac{\eta}{2} \sum_{t=1}^{T} \left( \|g_t\|_{\tilde{H}_t}^* \right)^2 \leq \frac{\eta\sqrt{r}}{2} \sum_{t=1}^{T} \left( \|g_t\|_{\hat{H}_t}^* \right)^2 \leq \eta\sqrt{r}\,\text{tr}(\hat{H}_T) \leq \eta r\,\text{tr}(\tilde{H}_T)\ .$$

Combining the bound on $R_D$ and $R_G$, the overall regret is

$$\text{Regret}_T \leq R_D + R_G \leq \left( \frac{D^2}{2\eta} + \eta r \right) \text{tr}(\tilde{H}_T) = \sqrt{2r}D\,\text{tr}(\tilde{H}_T) = \sqrt{2r}D\,\text{tr}(\tilde{L}_T^{1/4})\,\text{tr}(\tilde{R}_T^{1/4})\ .$$

by the choice of $\eta = \frac{D}{\sqrt{2r}}$ and trace multiplicative equality in Lemma 15. Finally, we have

$$\text{tr}\left( \tilde{L}_T^{1/4} \right) \leq_1 \text{tr}\left( \bar{L}_T^{1/4} \right) + \text{tr}\left( \left(\rho_{1:T}^L I_m\right)^{1/4} \right)$$

$$\leq_2 \text{tr}\left( \left( \sum_{t=1}^{T} G_t G_t^\top + \varepsilon I \right)^{1/4} \right) + m\left(\rho_{1:T}^L\right)^{1/4}$$

$$= \text{tr}\left( L_T^{1/4} \right) + m\left(\rho_{1:T}^L\right)^{1/4}\ ,$$

where $\leq_1$ follows from definition of $\tilde{L}_T$ in Algorithm 3 and subadditivity of $\text{tr}\left( (\cdot)^{1/4} \right)$ [47], $\leq_2$ follows from Remark 11. Similarly,

$$\text{tr}\left( \tilde{R}_T^{1/4} \right) \leq \text{tr}\left( R_T^{1/4} \right) + n\left(\rho_{1:T}^R\right)^{1/4}\ .$$

$\qquad\square$

### B.5.2 Proof of Lemma 14

*Proof.* We will show the first inequality as the second inequality holds analogously. For $t = 0$, $\tilde{L}_0 = \varepsilon I_m$ by definition of algorithm. Suppose the first inequality holds for $t$. Consider $t+1$:

$$\tilde{L}_{t+1} = \bar{L}_{t+1} + \rho_{1:t+1}^L I_m$$

$$\succeq_1 \bar{L}_t + G_{t+1} G_{t+1}^\top + \rho_{t+1}^L N_{t+1} + \rho_{1:t}^L I_m$$

$$= \tilde{L}_t + G_{t+1} G_{t+1}^\top + \rho_{t+1}^L N_{t+1}$$

$$\succeq_2 \sum_{s=1}^{t+1} G_s G_s^\top + \sum_{s=1}^{t+1} \rho_s^L N_s^L + \varepsilon I_m\ ,$$

where $\succeq_1$ follows from Observation 13 and $\succeq_2$ follows from induction hypothesis. $\qquad\square$

Table 4: The search space for hyperparameters for tuning Shampoo on our NN architectures for the Kronecker-factored covariance optimization. Note that the search space explores one less momentum, not momentum directly. Label smoothing was only applied to ImageNet. We sample uniformly either from linear or logscale among the ranges specified with 100 trials, and select the best one according to validation accuracy.

| Hyperparameter | Range | Log scale? |
|---|---|---|
| Learning rate $\eta$ | $[10^{-4}, 10^{-2}]$ | ✓ |
| Weight decay $\gamma$ | $[10^{-2}, 1]$ | ✓ |
| Momentum $1 - \beta_1$ | $[10^{-2}, 10^{-1}]$ | ✓ |
| Label smoothing | $[0, 0.2]$ | |

## C   Training Settings

For repeatable, standard evaluation on modern, competitive tasks we use `init2winit` [48] for Jax [49] implementations of architectures in Flax [50] and standard dataset preprocessing built on top of TFDS [51]. We rely on standard scientific packages for conducting our work [52, 53, 54, 55]. Our source code will be released after publication.

Neural net architecture settings are taken from the default settings of the `init2winit` library at hash `e337ffe` [48], which reference the MLCommons specifications provided at MLCommons® open engineering consortium [56], including the MLPerf ResNet-50 variant [57], Conformer, and GNN. The Distributed Shampoo implementation was run at hash `83e6e62` in the repository referenced by Anil et al. [9].

Throughout, weight decay is applied using its decoupled variant [58].

We requested a Shampoo tuning script from Dayma and Anil [59], Anil et al. [9], which fixed several parameters for Shampoo outside the usual defaults. We tuned on a cluster of TPUv4s, with 16 TPUv4s per trial in data-parallel mode.

- Block size was already set to 1024. As mentioned in Sec. 5.2, we kept this change for consistency in covariance factor size across architectures.
- Preconditioning was set to start 101 steps into training (`start_preconditioning_step`).
- Preconditioners were updated every 10 steps instead of every step for speed (`preconditioning_compute_steps` is 10).
- The grafting type, which controls the per-tensor learning rate schedule, was set to `RMSPROP_NORMALIZED`, which applies RMSProp [60] over unit-normalized gradients.
- `moving_average_for_momentum` was activated (so the final updates are computed as $\beta_1\mu_t + (1 - \beta_1)g_t$, where $\mu_t$ is the momentum term and $g_t$ is the preconditioned update.
- The virtual batch size, used to compute batch norm statistics, was set to 32 (the full per-step minibatch size was 1024, but this enables data-parallel training).

Also from the provided script, we used a linear warmup rampup starting from 0 to the nominal learning rate hyperparameter setting, followed by a cosine decay schedule, with the transition happening 5% of the way into training (the learning rate monotonically increases, then montonically decreases, as the cosine schedule has a quarter-period set to the number of training steps).

Then, we performed tuning using random hyperparameter search over the space defined in Tbl. 4. We ran the batch sizes and number of steps provided in the scripts, which were 256, 512, 1024 for Conformer, GNN, and ResNet-50, respectively, for about 162, 117, 199 epochs, respectively.

Shampoo is automatically configured with grafting parameters, which we search over [61].

## D   ResNet-50 Settings

For training ImageNet, we mostly inherited the settings of Appendix C for Shampoo tuning, but made some minor modifications, namely adding of the second moment decay ($\beta_2$), widening the search

Table 5: The search space for hyperparameters for tuning Shampoo on our architectures for ImageNet hyperparameters. The same space was applied to `S-Shampoo` with a fixed sketch rank $\ell = 256$ for all tensors. Note that we search $1 - \beta_1, 1 - \beta_2$, and not the original hyperparameter. We sample uniformly either from linear or logscale among the ranges specified with 256 trials, and select the best one according to validation accuracy. $(*)$ stands for a discrete uniform choice over four different grafting rates, based on AdaGrad, RMSProp, and normalized versions of the two. The gradient clipping norm is similarly discrete.

| Hyperparameter | Range | Log? |
|---|---|---|
| Learning rate $\eta$ | $[10^{-4}, 10^{-2}]$ | ✓ |
| Weight decay $\gamma$ | $[10^{-3}, 0.1]$ | ✓ |
| Momentum $1 - \beta_1$ | $[10^{-4}, 10^{-1}]$ | ✓ |
| 2$^{\text{nd}}$ moment $1 - \beta_2$ | $[10^{-4}, 10^{-1}]$ | ✓ |
| Label smoothing | $[0, 0.2]$ | |
| Dropout Rate | $[0, 0.2]$ | |
| Grafting Type | $(*)$ | |
| Gradient Clip $L_2$ | $\{1, 10, 10^2, 10^3\}$ | |

Table 6: The search space for hyperparameters for tuning Adam on our architectures for ImageNet hyperparameters. The same caveats as in Tbl. 5 apply. Also tuned with 256 trials.

| Hyperparameter | Range | Log? |
|---|---|---|
| Learning rate $\eta$ | $[10^{-4}, 10^{-2}]$ | ✓ |
| Weight decay $\gamma$ | $[10^{-3}, 0.1]$ | ✓ |
| Momentum $1 - \beta_1$ | $[10^{-4}, 10^{-1}]$ | ✓ |
| 2$^{\text{nd}}$ moment $1 - \beta_2$ | $[10^{-4}, 10^{-1}]$ | ✓ |
| Label smoothing | $[0, 0.2]$ | |
| Dropout Rate | $[0, 0.2]$ | |
| Warmup Duration | $[2\%, 10\%]$ of training | |
| Gradient Clip $L_2$ | $\{1, 10, 10^2, 10^3\}$ | |

space, and, for computational reasons, performing a shortened run of only 66 epochs for tuning trials. The architecture details remain the same. The learning rate schedule was stretched to this interval, so warmup was still 5% of the duration, and cosine decay ended learning rate at 0 by the end of the 66 epochs of training. The full search space is elaborated on in Tbl. 5.

To tune Adam, a first order method, we considered mostly the same nominaly hyperparameters (where $\beta_2$ refers to second moment momentum now), except grafting, which instead we replaced with a search over the warmup duration, summarized in Tbl. 6.

Full evaluation of the selected best hyperparameters for each setting was performed with the classical 90-epoch setting, with the learning rate schedule correspondingly stretched.

We provide the full training curves in Fig. 5.

# E Conformer Settings

The Conformer architecture was used from the MLCommons specification as described in Appendix C, with the following fixed additional settings: 1024 batch size, 100 epochs of training, 5% of training used for linear warmup with cosine decay of learning rate. We fixed gradient clipping at a value of 10, without which we noticed Adam curves were very volatile. We set the `eigh` parameter for Shampoo to true (we found that it did not make a difference in a few sample runs' loss curves, as an alternative to the iterative $p$-th inverse root routine in Shampoo, but used it instead since we believe it has better numerical stability). The hyperparameters we searched over for all optimizers are described in Tbl. 7.

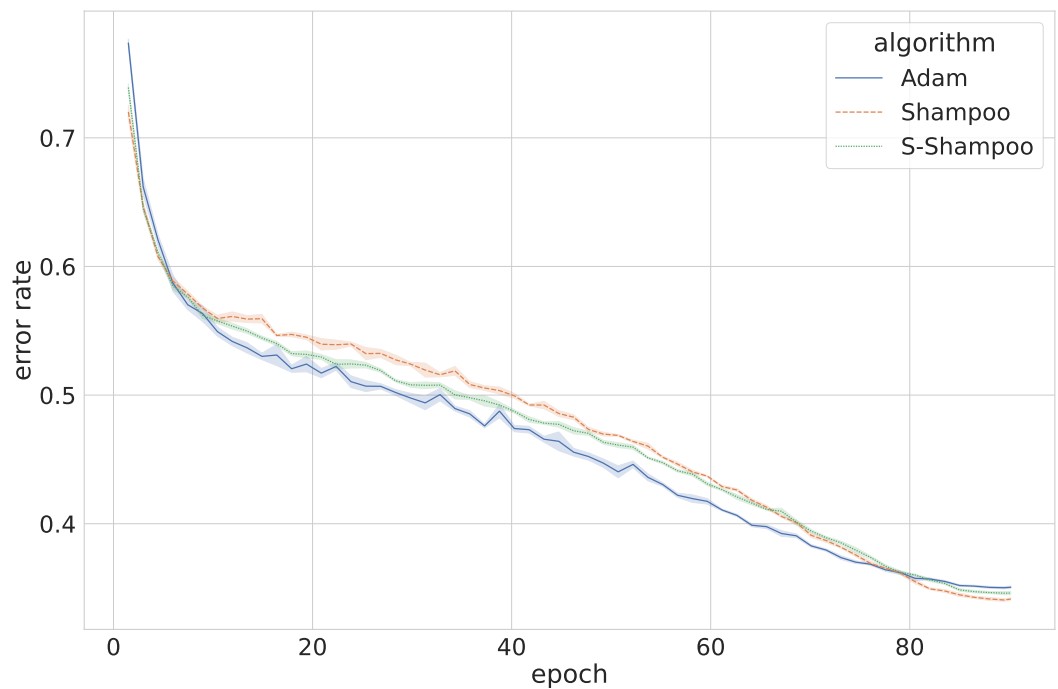

Figure 5: Full test set curves for imagenet.

Table 7: The search space for hyperparameters for tuning Shampoo, Adam, and `S-Shampoo` for Conformer with fixed 256 rank. Here we fixed the grafting type to be RMSProp. During initial runs of the baselines, we noticed that Adam preferred larger learning rates, so we changed and reran its space for $\eta$ to be $10\times$ that of Shampoo, namely $[10^{-4}, 10^{-2}]$, still searching over logspace. We also stopped any hyperparameter trial which did not go below 0.875 WER after 5000 training steps.

| Hyperparameter | Range | Log? |
|---|---|---|
| Learning rate $\eta$ | $[10^{-5}, 10^{-3}]$ | ✓ |
| Momentum $1 - \beta_1$ | $[10^{-3}, 10^{-1}]$ | ✓ |
| 2nd moment $1 - \beta_2$ | $[10^{-3}, 10^{-1}]$ | ✓ |
| Weight decay $\gamma$ | $[10^{-4}, 10^{-2}]$ | ✓ |
| Dropout Rate | $[0, 0.2]$ | ✓ |

## F   GNN Settings

The GNN architecture was used from the MLCommons specification as described in Appendix C, with the following fixed additional settings: 1024 batch size, 30 epochs of training, 5% of training used for linear warmup with cosine decay of learning rate, 0.05 dropout, and we set the `eigh` parameter for Shampoo to true as in Appendix E.

Then we searched a hyperparameter space for Shampoo, Adam, and `S-Shampoo` as described in Tbl. 8.

Table 8: The search space for hyperparameters for tuning Shampoo and `S-Shampoo` for GNN with fixed regularization settings (due to resource constraints, we only ran 128 samples from the grid here). Here we fixed the grafting type to be RMSProp and did not use gradient clipping, unlike Tbl. 5, based on a few trial runs of the Shampoo baseline from which we determined we could reduce the hyperparameter space.

| Hyperparameter | Range | Log? |
|---|---|---|
| Learning rate $\eta$ | $[10^{-4}, 10^{-2}]$ | ✓ |
| Momentum $1 - \beta_1$ | $[10^{-3}, 0.5]$ | ✓ |
| 2$^{\text{nd}}$ moment $1 - \beta_2$ | $[10^{-3}, 0.5]$ | ✓ |
| Weight decay $\gamma$ | $[10^{-3}, 0.5]$ | ✓ |

# G    Step-skipping

In this section, we provide some theoretical justification for step-skipping. We first derive the regret bound of AdaGrad with step skipping, named Generic Epoch AdaGrad. The additional regret incurred is expressed as an error term. Then, we describe a setting where the error term admits a simple bound, showing that step-skipping incurs at most an extra $\log T$ time dependence on the regret.

## G.1    Adversarial losses

Consider a generalized epoching AdaGrad with $K$ fixed update points $t_k$, such that $t_1 = 0$ and $t_K = T$.

---
**Algorithm 5** Generic Epoch AdaGrad
---

1: **Input:** $\eta, T, \{t_k\}_{k=1}^K, G_0 \succ 0$, convex closed set $\mathcal{K}$.
2: **Initialize:** $x_1$.
3: **for** $k = 1, \dots, K - 1$ **do**
4:     **for** $t = t_k + 1, \cdots, t_{k+1}$ **do**
5:         Play $x_t$, receive $f_t$ loss with gradient $g_t$.
6:         Update $G_t = G_{t-1} + g_t g_t^\top$.
7:         Update $x_{t+1} = \Pi_{\mathcal{K}}[x_t - \eta G_{t_k}^{-1/2} g_t]$.
8:     **end for**
9: **end for**

---

**Theorem 18.** *Generic Epoch AdaGrad (Alg. 5) with fixed update points $\{t_k\}_{k=1}^K$ satisfies*

$$R_T \le \frac{D^2}{\eta} \operatorname{tr} G_T^{1/2} + \frac{\eta}{2} \left( 2 \operatorname{tr} G_T^{1/2} - 2 \operatorname{tr} G_0^{1/2} + \sum_{k=1}^K \epsilon_k \right) ,$$

*where the error terms $\epsilon_k$ are given by*

$$\epsilon_k = \operatorname{tr} \left( G_{t_k}^{-1/2} S_k G_{t_k}^{-1/2} A_k \right) ,$$
$$S_k = \int_0^\infty \exp \left( -\tau G_{t_k}^{1/2} \right) A_k \exp \left( -\tau G_{t_k}^{1/2} \right) \mathrm{d}\tau ,$$
$$A_k = G_{t_{k+1}} - G_{t_k} .$$

*Proof of Theorem 18.* First we start with the usual decomposition.

**Lemma 19.** *Consider arbitrary adversarial convex losses $f_t$. Without projection, the regret $R_T$ relative to a comparator $x_*$ with $D = \max_t \|x_t - x_*\|_2$, for generic epoch AdaGrad with fixed update points $t_k$ is given by*

$$R_T \le \frac{D^2}{\eta} \operatorname{tr} G_T^{1/2} + \frac{\eta}{2} \sum_{k=1}^K \sum_{t=t_k+1}^{t_{k+1}} g_t^\top G_{t_k}^{-1/2} g_t .$$

*Proof of Lemma 19.* This follows from the usual AdaGrad analysis since our preconditioners are monotone $G_{t_k} \succeq G_{t_{k+1}}$. □

So we must turn our attention to the gradient bound. We start by noting the following lemmas established in matrix analysis.

**Lemma 20** (Corollary 4.1 in [62]). *The map $f(X) = X^{-1/2}$ is matrix convex over the positive definite domain; i.e., for any two matrices $A, B \succ 0$ and any $\theta \in [0, 1]$, we have*

$$\theta f(A) + (1 - \theta) f(B) \succeq f(\theta A + (1 - \theta) B) .$$

**Lemma 21** (Theorem V.3.3, Exercise V.3.15 in [63]). *Suppose a matrix convex function $F(X)$ is induced by applying $f$ pointwise to its spectrum $F(X) = U \operatorname{diag}[f(\Lambda_{ii})]U^\dagger$ with $f \in \mathcal{C}^1(I)$ for some $I \subset \mathbb{R}_+$. Then*

$$F(X) + \partial F(X)(\Delta) \preceq F(X + \Delta) \ ,$$

*if and only if $F(X)$ is matrix convex, and the linear transformation $\partial F(X)$ is the derivative of $F$ at $X$.*

Matrix derivative computation [64, 65] shows that if $F(X) = X^{-1/2}$ then

$$\partial F(X)(\Delta) = -X^{-1/2}\left[(X^{1/2} \oplus X^{1/2})^{-1}\Delta\right]X^{-1/2} \ ,$$

where $(X^{1/2} \oplus X^{1/2})^{-1}\Delta$ is the solution $S$ to the continuous Lyapunov equation $\sqrt{X}S + S\sqrt{X} = \Delta$ as $X \oplus X = I \otimes X + X \otimes I$. For $X \succ 0$, it is known from generic results about Sylvester's equation that the solution $S$ is unique. Since $-X$ is asymptotically stable in the Lyapunov sense,

$$S(X, \Delta) = \int_0^\infty \exp(-\tau\sqrt{X})\Delta\exp(-\tau\sqrt{X})\,\mathrm{d}\tau \ .$$

With these results from matrix analysis and linear systems, we are ready to bound the gradient term in Lemma 19. Consider a single term from the gradient bound in Lemma 19, $\sum_{t=t_k+1}^{t_{k+1}} g_t^\top G_{t_k}^{-1/2} g_t$ for fixed $k$.

With $X = G_{t_k}$, $\Delta = A_k = G_{t_{k+1}} - G_{t_k}$, and $f(X) = X^{-1/2}$ consider applying Lemma 21. $F(X) \preceq F(X + A_k) - \partial F(X)(A_k)$, so overall

$$\sum_{t=t_k+1}^{t_{k+1}} g_t^\top G_{t_k}^{-1/2} g_t$$

$$\leq \sum_{t=t_k+1}^{t_{k+1}} g_t^\top \left[G_{t_{k+1}}^{-1/2} - \partial F(G_{t_k})(A_k)\right] g_t$$

$$= \sum_{t=t_k+1}^{t_{k+1}} g_t^\top G_{t_{k+1}}^{-1/2} g_t - \operatorname{tr}\left(\partial F(G_{t_k})(A_k) \sum_{t=t_k+1}^{t_{k+1}} g_t g_t^\top\right)$$

$$= \sum_{t=t_k+1}^{t_{k+1}} g_t^\top G_{t_{k+1}}^{-1/2} g_t - \operatorname{tr}\left(\partial F(G_{t_k})(A_k)A_k\right)$$

$$= \sum_{t=t_k+1}^{t_{k+1}} g_t^\top G_{t_{k+1}}^{-1/2} g_t + \operatorname{tr}\left(G_{t_k}^{-1/2} S_k G_{t_k}^{-1/2} A_k\right)$$

$$= \sum_{t=t_k+1}^{t_{k+1}} g_t^\top G_{t_{k+1}}^{-1/2} g_t + \epsilon_k \ .$$

**Lemma 22** (FTL-BTL with errors). *Consider arbitrary $\phi_k$ for $k \in 0, 1, \cdots, K$. Let $x_k \in \operatorname{argmin} \sum_{j=0}^k \phi_j$ and suppose $\phi_k(x_{k-1}) \leq \phi_k(x_k) + \delta_k$. Then $\forall K$,*

$$\sum_{k=0}^K \phi_k(x_{k-1}) \leq \sum_{k=0}^K \phi_k(x_K) + \delta_k \ ,$$

*with $\delta_0 = 0$ and $x_{-1} = x_0$.*

Assume Lemma 22 and take $\phi_k(X) = \langle A_k, X\rangle$ for $X \succeq 0$ and $\phi_0(X) = \langle G_0, X\rangle + \operatorname{tr} X^{-1}$. Note that

$$\sum_{j=0}^k \phi_j(X) = \operatorname{tr} X^{-1} + \sum_{j=0}^k \langle A_j, X\rangle \ .$$

In particular, $G_{t_{k+1}}^{-1/2} = \operatorname{argmin}_{X \succeq 0} \sum_{j=0}^{k} \phi_j(X)$.

Furthermore, with $\delta_k = \epsilon_k$, the condition $\phi_k(G_{t_k}) \leq \phi_k(G_{t_{k+1}}) + \delta_k$ is satisfied. Lemma 22 implies

$$\sum_{k=1}^{K} \sum_{t=t_k+1}^{t_{k+1}} g_t^\top G_{t_k}^{-1/2} g_t = \sum_{k=1}^{K-1} \phi_k\left(G_{t_k}^{-1/2}\right)$$

$$= -\phi_0(G_0^{-1/2}) + \sum_{k=0}^{K-1} \phi_k\left(G_{t_k}^{-1/2}\right)$$

$$\leq -\phi_0(G_0^{-1/2}) + \sum_{k=0}^{K-1} \phi_k\left(G_T^{-1/2}\right) + \epsilon_k \ ,$$

where $G_{t_0} \stackrel{\text{def}}{=} G_0$. Lastly, since

$$\sum_{k=0}^{K-1} \phi_k\left(G_T^{-1/2}\right) = \operatorname{tr} G_T^{1/2} + \operatorname{tr}\left(G_T^{-1/2} \sum_{k=0}^{K-1} A_k\right) = 2\operatorname{tr} G_T^{1/2},$$

we conclude with the desired bound for $R_T$. $\qquad\square$

*Proof of Lemma 22.* By induction on $K$. For $K = 0$, $\phi_0(x_{-1}) = \phi_0(x_0)$, holding by definition. Suppose that the hypothesis now holds for $K$; it then holds for $K + 1$.

$$\sum_{k=0}^{K+1} \phi_k(x_{K+1}) = \sum_{k=0}^{K} \phi_k(x_{K+1}) + \phi_{K+1}(x_{K+1})$$

$$\geq \phi_{K+1}(x_{K+1}) + \sum_{k=0}^{K} \phi_k(x_K)$$

$$\geq \phi_{K+1}(x_K) - \epsilon_{K+1} + \sum_{k=0}^{K} \phi_k(x_K)$$

$$\geq \phi_{K+1}(x_K) - \epsilon_{K+1} + \sum_{k=0}^{K} \phi_k(x_{k-1}) - \epsilon_k$$

$$\geq \sum_{k=0}^{K+1} \phi_k(x_{k-1}) - \epsilon_k \ .$$

$$\square$$

## G.2   Simplifying the error

We want to simplify the term $\epsilon_k$, which is given by

$$\epsilon_k = \operatorname{tr}\left(G_{t_k}^{-1/2} S_k G_{t_k}^{-1/2} A_k\right) \ ,$$

$$S_k = \int_0^\infty \exp\left(-\tau G_{t_k}^{1/2}\right) A_k \exp\left(-\tau G_{t_k}^{1/2}\right) d\tau \ ,$$

$$A_k = G_{t_{k+1}} - G_{t_k} \ .$$

Next, notice that $X$ and $\exp\left(-\alpha X^{-1}\right)$ commute. Then along with linearity of trace, we can established that

$$\epsilon_k = \int_0^\infty \operatorname{tr}\left[\left(\exp\left(-\tau G_{t_k}^{1/2}\right) G_{t_k}^{-1/2} A_k\right)^2\right] d\tau \ .$$

### G.3 Towards simpler error

$\epsilon_k$ can be further simplified and bounded under additional assumptions. Namely,

**Assumption 1.** *Suppose that w.p. at least $1 - \delta/2K$ for some fixed, universal $\beta > 0$, we have the inequality $A_k \preceq \beta G_{t_k}$ where $A_k \stackrel{def}{=} G_{t_{k+1}} - G_{t_k}$.*

**Assumption 2.** *Suppose that w.p. at least $1 - \delta/2K$, $G_{t_k}$'s are $(\sigma_{\min}, \sigma_{\max})$-well-conditioned, i.e.*

$$\lambda_d(G_{t_k}) \geq \sigma_{\min} t_k \quad and \quad \lambda_1(G_{t_k}) \leq \sigma_{\max} t_k.$$

**Remark 23.** As an example, consider the stochastic linear setting where at each iteration we receive a loss function $\langle g_t, x \rangle$, and $g_t$'s are independent, though not necessarily identically distributed, and satisfies that $2\sigma_{\min} I \preceq \mathbb{E}[g_t g_t^\top] \preceq \frac{\sigma_{\max}}{2} I$ and $\|g_t\|_2 \leq \sqrt{\frac{\sigma_{\max}}{2}}$ almost surely. Then, for $T$ sufficiently large and $t_{k+1} - t_k = O(\log T)$, by matrix Chernoff bounds Assumption 1 and 2 are satisfied.

First, $\forall X, Y \succeq 0$, the following inequality hold:

**Lemma 24.** *If $X \preceq Y$ and $A \succeq 0$, then $\operatorname{tr}\left[(AX)^2\right] \leq \operatorname{tr}\left[(AY)^2\right]$.*

With Lemma 24, we can bound $\epsilon_k$. With probability at least $1 - \delta/2K$,

$$\begin{aligned}
\epsilon_k &= \int_0^\infty \operatorname{tr}\left[\left(\exp\left(-\tau G_{t_k}^{1/2}\right) G_{t_k}^{-1/2} A_k\right)^2\right] \mathrm{d}\tau \\
&\leq \beta^2 \int_0^\infty \operatorname{tr}\left[\left(\exp\left(-\tau G_{t_k}^{1/2}\right) G_{t_k}^{-1/2} G_{t_k}\right)^2\right] \mathrm{d}\tau \\
&= \beta^2 \int_0^\infty \operatorname{tr}\left[\left(\exp\left(-\tau G_{t_k}^{1/2}\right) G_{t_k}^{1/2}\right)^2\right] \mathrm{d}\tau \\
&= \beta^2 \int_0^\infty \operatorname{tr}\left(\exp\left(-2\tau G_{t_k}^{1/2}\right) G_{t_k}\right) \mathrm{d}\tau ,
\end{aligned}$$

where the last step only holds since $X$ and $\exp\left(-\alpha X^{-1/2}\right)$ commute.

Next, let $\lambda_i$ denote the $i$-th largest eigenvalue and $\lambda_{-i}$ be the $i$-th smallest. Notice since $\exp\left(-\tau G_{t_k}^{1/2}\right)$, $G_{t_k}^{1/2}$, and $G_{t_k}$ are simultaneously diagonalizable, and montonic matrix functions preserve eigenvalue ordering, we have

$$\begin{aligned}
\lambda_i\left(\exp\left(-2\tau G_{t_k}^{1/2}\right)\right) &= \exp\left(-2\tau \lambda_{-i}\left((G_{t_k})^{1/2}\right)\right) , \\
\lambda_i\left(\exp\left(-2\tau G_{t_k}^{1/2}\right) G_{t_k}\right) &= \lambda_i\left(\exp\left(-2\tau G_{t_k}^{1/2}\right)\right) \lambda_i(G_{t_k}) .
\end{aligned}$$

Returning to our $\epsilon_k$ bound, rewriting the trace with eigenvalues, we have w.p. at least $1 - \delta/2K$,

$$\begin{aligned}
\epsilon_k &\leq_1 \beta^2 \int_0^\infty \sum_i \lambda_i\left(\exp\left(-2\tau G_{t_k}^{1/2}\right) G_{t_k}\right) \mathrm{d}\tau \\
&= \beta^2 \sum_i \lambda_i(G_{t_k}) \int_0^\infty \exp\left(-2\tau \lambda_{-i}(G_{t_k})^{1/2}\right) \mathrm{d}\tau \\
&= \beta^2 \sum_i \frac{\lambda_i(G_{t_k})}{2\lambda_{-i}(G_{t_k})^{1/2}} ,
\end{aligned}$$

where $\leq_1$ follows from Tonelli's Theorem. At this point, we apply Assumption 2 and get that w.p. at least $1 - \delta/K$,

$$\epsilon_k \leq \frac{\beta^2}{\sqrt{t_k \sigma_{\min}}} \sum_i \lambda_i(G_{t_k})$$

$$\leq \frac{\beta^2}{\sqrt{t_k \sigma_{\min}}} \sum_i (t_k \sigma_{\max})^{1/2} \lambda_i(G_{t_k})^{1/2}$$

$$= \beta^2 \sqrt{\frac{\sigma_{\max}}{\sigma_{\min}}} \sum_i \lambda_i(G_{t_k})^{1/2}$$

$$= \beta^2 \sqrt{\frac{\sigma_{\max}}{\sigma_{\min}}} \operatorname{tr} G_{t_k}^{1/2} \; .$$

Across all epochs, we then have w.p. at least $1 - \delta$,

$$\frac{1}{\beta^2} \sqrt{\frac{\sigma_{\min}}{\sigma_{\max}}} \sum_{k=1}^K \epsilon_k \leq \sum_{k=1}^K \operatorname{tr} G_{t_k}^{1/2} \leq \log T \operatorname{tr} G_T^{1/2} \; .$$

Altogether, since $\beta$ is a universal constant, w.p. at least $1 - \delta$,

$$R_T \lesssim \frac{D^2}{\eta} \operatorname{tr} G_T^{1/2} + \eta \sqrt{\frac{\sigma_{\max}}{\sigma_{\min}}} \log T \operatorname{tr} G_T^{1/2} \; .$$

We conclude that in this case, the time dependency of Epoch AdaGrad's regret is only $\log T$ factor worse than that of the original AdaGrad regret.

### G.3.1  Proof of Lemma 24

First, for $0 \preceq X \preceq Y$, $BXB \preceq BYB$, $\forall B$, since $(Bx)^\top (Y - X)(Bx) \geq 0$, $\forall x$. By cyclic property of trace and taking $B = A^{1/2}X$,

$$\operatorname{tr}(AXAX) = \operatorname{tr}\left(X^{1/2}AXAX^{1/2}\right) \leq \operatorname{tr}\left(X^{1/2}AYAX^{1/2}\right) \; .$$

Continuing,

$$\operatorname{tr}\left[(AX)^2\right] \leq \operatorname{tr}\left(X^{1/2}AYAX^{1/2}\right)$$

$$= \operatorname{tr}\left(Y^{1/2}AXAY^{1/2}\right)$$

$$\leq \operatorname{tr}\left(Y^{1/2}AYAY^{1/2}\right)$$

$$= \operatorname{tr}\left[(AY)^2\right] \; .$$