# OpenReview forum: "Sketchy: Memory-efficient Adaptive Regularization with Frequent Directions"
_NeurIPS.cc/2023/Conference — NeurIPS 2023 poster_

### Official Review · Reviewer_6Jc9 · 2023-07-05

**Soundness:** 3 good
**Presentation:** 3 good
**Contribution:** 2 fair
**Rating:** 6
**Confidence:** 3

**Summary:**

This paper looks at using Frequent-Directions to do (non-diagonal) adagrad more efficiently.  It suggests doing this for both full-matrix adagrad and Shampoo, a version that takes into account the matrix structure of the parameters.

**Strengths:**

I think it's a pretty reasonable and natural idea, and both the theory and experiments
seem somewhat promising.

**Weaknesses:**

- The experiments don't get any actual improvement in time or memory relative to existing implementations

- The theorems seem a bit weak.  Take Theorem 3; in the case of no or limited spectral decay, the second error term (with Omega_L) is about d/sqrt(l) times bigger than the first term -- so it doesn't reduce to the first term, even for l=d/2 when it seems like it should be possible.

  If l is (d/10), as suggested by the experiments, isn't that still a sqrt(d) factor increase in the loss?  This seems poor... so poor relative to the experiments that I suspect that the true bound shouldn't have that extra sqrt(d).


Basically my concern with this paper is: the idea of using FD for this problem doesn't seem that innovative, which would be fine if this paper really demonstrated that FD is a good idea... but having read this paper, I'm still not confident that FD *is* a good approach here.  It's a reasonable thing to try, certainly, but that was already true before reading the paper.

**Questions:**

- Line 93 says that FD is "optimal up to universal multiplicative   constants," but IMO that's a bit of an overstatement -- it's minimax   optimal for its particular guarantee, but you could either (1) want  a different guarantee, or (2) want a guarantee for a distribution that isn't that particular hard instance.  In particular, it might well not be optimal up to constants for the instances that appear in this paper.

- Is Section 5.1 using beta_2 = 0.999?  It never says that it is, but the rest of the paper implies that it should.


**Limitations:**

Yes.

---

> ### Author Rebuttal · Authors · 2023-08-08
>
> Thank you for taking the time and effort to read and review our paper! Here are the responses to your concerns:
>
> Weaknesses:
>
> 1. The experiments don't get any actual improvement in time or memory relative to existing implementations.
>
> **Response**: Please see the top comment regarding performance numbers.
>
> 2. The theorems seem a bit weak. Take Theorem 3; in the case of no or limited spectral decay, the second error term (with Omega_L) is about d/sqrt(l) times bigger than the first term -- so it doesn't reduce to the first term, even for l=d/2 when it seems like it should be possible. If l is (d/10), as suggested by the experiments, isn't that still a sqrt(d) factor increase in the loss? This seems poor... so poor relative to the experiments that I suspect that the true bound shouldn't have that extra sqrt(d).
>
> **Response**: We have a tighter bound in Corollary 4. We agree that this bound is not tight, but we want to point out that previous works have shown for second-order methods superlinear memory is required (see the paper "Efficient Convex Optimization Requires Superlinear Memory" by Marsden et. al.). In our case, the guarantee is established for our proposed algorithm which requires near linear memory $O(mk+nk)$. More importantly, the performance of our algorithms depends on the adaptivity of the regret bound from the term $\Omega_\ell$, which we showed empirical evidence for in Figure 2.
>
> Basically my concern with this paper is: the idea of using FD for this problem doesn't seem that innovative, which would be fine if this paper really demonstrated that FD is a good idea... but having read this paper, I'm still not confident that FD is a good approach here. It's a reasonable thing to try, certainly, but that was already true before reading the paper.
>
> The set of “ideas to try'” may have included FD-for-optimization before this paper. But our work shows that a much more specified, concrete variant, the composition of FD with Shampoo is rigorously-defined belongs to a much narrower set of optimization algorithms, those which are theoretically and practically validated to provide low generalization error (for NNs) or regret (for convex online learners).
>
> This required a novel integration between FD, Shampoo, and their regret analysis. Previous sketching work was never applied to large neural networks (even empirically focussed ones, like GGT, limit themselves to small resnets). We hope that the shift to approximate second order methods on the KF covariance factors, rather than full per-layer covariance, spur further interest in this algorithmic direction, including further development of efficient implementations.  those which require sublinear memory for accumulators, which were shown on large neural benchmarks such as ResNet-50 on imagenet to actually work better than Adam (unlike all previous low-memory works), and which have a path to an efficient implementation (with iterative top-k eigenvalue routines instead of SVD).
>
> Questions:
>
> 1. Line 93 says that FD is "optimal up to universal multiplicative constants," but IMO that's a bit of an overstatement -- it's minimax optimal for its particular guarantee, but you could either (1) want a different guarantee, or (2) want a guarantee for a distribution that isn't that particular hard instance. In particular, it might well not be optimal up to constants for the instances that appear in this paper.
>
> **Response**: This is a fair catch. We’ve updated our language.
>
> 2. Is Section 5.1 using beta_2 = 0.999? It never says that it is, but the rest of the paper implies that it should.
>
> **Response**: $1 - 10^{-4}$ is the largest $\beta_2$ in the hyperparameter grid (for all optimizers & all neural nets); full grids in the appendices. Using lower $\beta_2$ values (which in practice are selected by the hyperparameter search) results in even more dramatic spectral decay. For instance, for resnet shampoo, the optimal selected $\beta_2$ was actually 0.988. Repeating the synthetic data experiment on L272 to L274, the covariance factor’s intrinsic rank for random normal gradients falls from 862.13 (0.25) to 585.30 (0.45) as $\beta_2$ goes from 0.999 to 0.988.

---

> > ### Comment · Reviewer_6Jc9 · 2023-08-12
> >
> > Thanks for your response.  I'm still a little concerned that the theorem seems loose and the experiment isn't very comprehensive, but your point about validating a very concrete proposal is well taken.  I'm willing to raise my score to 6.

---

### Official Review · Reviewer_CbL3 · 2023-07-05

**Soundness:** 3 good
**Presentation:** 3 good
**Contribution:** 3 good
**Rating:** 6
**Confidence:** 4

**Summary:**

Motivated by empirical observation that Kronecker-factored gradient covariance matrix is (approximately), this work proposed a sketching scheme for memory-efficient preconditioner for gradient-based optimizer.

**Strengths:**

- The observation of the low-rankedness of the KF convariance matrix in Figure 3 is quite intriguing.
- The regret analysis of the sketchy for OCO looks sound.

**Weaknesses:**

- From Figure 1, it seems that the main application of sketchy is S-Shampoo that achieves the sublinear memory. However, I wonder for shampoo (where the factorization has a certain structure) doing a sketching according to sketchy would actually lead to a low-rank KF covariance matrix? For the adagrad case, it makes a complete sense that Figure 3 provides an empirical justification for S-Adagrad, but I'm quite not sure whether this actually justifies S-Shampoo.

-Also, given the factorization structure of Shampoo, I'm not sure how doing a SVD and sketching individual factor leads to capturing the major component of the entire KF covariance matrix.

- Although the authors said they didn't focus on the implementation of S-Shampoo, it would be helpful for readers how the optimizers compare in terms of wall-time to see how sketchy fare in practice.


**Questions:**

- Although it's standard, it would be helpful for readers to summarize how the regret guarantee can be translated into the stationary point guarantee for nonconvex problems. Maybe, include a corollary/proposition so that readers can understand how the guarantees in this paper translate into the nonconvex land?

---

> ### Author Rebuttal · Authors · 2023-08-08
>
> Thank you for taking the time and effort to review our paper. Here are our responses to your concerns:
>
> Weaknesses:
>
> 1. From Figure 1, it seems that the main application of sketchy is S-Shampoo that achieves the sublinear memory. However, I wonder for shampoo (where the factorization has a certain structure) doing a sketching according to sketchy would actually lead to a low-rank KF covariance matrix? For the adagrad case, it makes a complete sense that Figure 3 provides an empirical justification for S-Adagrad, but I'm quite not sure whether this actually justifies S-Shampoo.
>
> **Response**: Figure 3 is plotting the intrinsic trace of the Kronecker factors (left and right) from Shamoo, individually. I.e., both factors simultaneously have fast spectral decay. Due to the nature of the way the Kronecker product composes eigensystems, this does also imply that the resulting overall matrix has a certain spectral decay as well, but because the total matrix dimension prohibitively large, S-Adagrad wouldn’t be practical to implement for DNNs.
>
> 2. Also, given the factorization structure of Shampoo, I'm not sure how doing a SVD and sketching individual factor leads to capturing the major component of the entire KF covariance matrix.
>
> **Response**: This is a great question, and gets at the harmony of why sketching composes well with Shampoo. It’s not required to approximate the entire KF covariance matrix well to have good preconditioner – thanks to Thm. 5, the regret from S-Shampoo scales with the individual spectral terms of each kronecker factor, not the overall one. This is due to the fact that the regret is bounded by the product of the trace of individual KF terms, rather than the entire matrix, in the Thm. 5 proof, ultimately from the inequality on L638 in the supplement.
>
> 3. Although the authors said they didn't focus on the implementation of S-Shampoo, it would be helpful for readers how the optimizers compare in terms of wall-time to see how sketchy fare in practice.
>
> **Response**: Please see the top-level comment, where we add wall-time numbers.
>
> Questions:
>
> We are happy to include a corollary in the revision to translate our regret bound to non-convex optimization. In summary, given a non-convex optimization problem $\min_x f(x)$, where $f$ is $L$-smooth and bounded, one can reduce it to solving strongly convex subproblems. Consider the following algorithm:
>
> For $t=1, ..., T$,
>
>   $x_{t+1} = \text{argmin}_x f(x) + 2L ||x - x_t||^2$
>
> Output $x_{t^*} = \text{argmin}_x ||\nabla f(x_t)||$.
>
> This algorithm is guaranteed to output an $\epsilon$-stationary point after $O(1/\epsilon^2)$ iterations. Note that each iteration is a convex optimization problem that can be solved with our algorithm, and hence it will translate into a nonconvex optimization guarantee. For stochastic non-convex optimization, the full algorithm requires $O(\mu^2\sigma^2/\epsilon^4)$ calls to the gradient oracle, where $\sigma^2$ is the variance of the stochastic gradient oracle, and $\mu$ is the ratio of adaptivity defined in [1].
>
> [1] Efficient Full-Matrix Adaptive Regularization. Naman Agarwal, Brian Bullins, Xinyi Chen, Elad Hazan, Karan Singh, Cyril Zhang, Yi Zhang Proceedings of the 36th International Conference on Machine Learning, PMLR 97:102-110, 2019.

---

> > ### Comment · Reviewer_CbL3 · 2023-08-13
> > **Thanks!**
> >
> > I read the response and it's satisfactory so I raise my score to 6.

---

### Official Review · Reviewer_fXUQ · 2023-07-06

**Soundness:** 3 good
**Presentation:** 2 fair
**Contribution:** 2 fair
**Rating:** 3
**Confidence:** 4

**Summary:**

This paper proposed a sketched online optimization method. The main idea is using frequent directions (FD) with adaptive regularization to maintain a preconditioner in the training process, which leads to $O(dk)$ memory and additive error associated with the small eigenvalues of the gradient covariance.

**Strengths:**

1. The design of the algorithm is well-motived and rigorous theoretical analysis is provided.
2. The experiments on deep nerul network shows the proposed algorithm improved the recent proposed baseline methods Shampoo.


**Weaknesses:**

My main consideration is the novelty of this paper. The relationship between proposed Sketchy AdaGrad (S-AdaGrad) and existing work [27, 43] is unclear. Here are the detailed comments:

1.	To my understanding, the main idea of S-AdaGrad (Algorithm 2) is using adaptive regularized frequent directions to estimate the gradient covariance. However, this idea is not novel. Luo et al. [43] proposed robust frequent directions (RFD) and applied it to online optimization. The main idea of RFD is also introducing an adaptive regularization, which is very similar to FD-update (Algorithm 1) in this paper. RFD uses $(\rho_{1:t} I)/2$ as the regularization term (rather than $\rho_{1:t} I$ in this paper). It is shown that this strategy has tighter error bound and is more well-conditioned than standard FD, while the results of Observation 6 looks no better than FD.
2.	The FD-update (Algorithm 1) looks the same as spectral compensation frequent directions (SCFD) [27]. Since sketched convex online optimization has been studied by Wan and Zhang [24], Luo et al. [25] and Luo et al. [43], the design of S-AdaGrad seems only applying a different sketching method SCFD to convex online optimization.


**Questions:**

1.	Can you provide a detailed discussion to compare the proposed algorithms with RFD, SCFD and related online optimization algorithms?
2.	SCFD is designed for linear contextual bandits and the analysis shows its positive definite monotonicity is important to establish the regret bound of this problem. In the scenario of convex online optimization, is there any theoretical result to show using SCFD is better than FD and RFD?


**Limitations:**

No potential negative societal impact. The detailed discussion for related work is necessary. Please see above for details.

---

> ### Author Rebuttal · Authors · 2023-08-08
>
> Thank you for taking the time and effort to review our paper. Here are our responses to your concerns:
>
> Weaknesses:
>
> RFD has the same regret bound (though different constants) as FD-SON, which as per Table 1, page 3 can be linear in T when there is no spectral decay or curvature (see our response to rVs5). More importantly, RFD gives up the approximate isometry between its preconditioner approximation and the true gradient covariance (see Remark 9 in the Appendix). This is precisely because of their choice to tighten the Frobenius norm by giving up the property that their preconditioner dominates the true covariance. However, this isometry is required in our analysis which demonstrates additive spectral error. The RFD paper essentially re-analyzes the FD-SON proof, and as such inherits its linear-regret bound in the no-spectral-decay case. We updated our related work to emphasize this.
>
>
> Questions:
>
> 1. Please see our top-level official comment. To summarize, all of the approaches you have mentioned have two fundamental deficiencies which our work addresses: (a) They require superlinear $O(mnr)$ memory, versus Sketchy’s $O(mk + nk)$, and (b) even for the S-Adagrad case, prior work does not provide a worst-case $O(\sqrt{T})$ regret bound when no spectral decay is present $(\Omega_k = \Theta(T))$. We present the first analysis which achieves this and demonstrate how this guarantee can rigorously compose with Shampoo.
>
> 2. The entire tree of citation descendants for SCFD has 9 papers according to Google Scholar. None focus on the general OCO case (in fact, all focus on linear bandits or bandits in general). Perhaps the reviewer has an analysis of SCFD for the general OCO case in mind and could provide a citation?

---

> > ### Comment · Reviewer_fXUQ · 2023-08-21
> >
> > Thanks for your response. I think the sketching method in proposed method is very similar to existing regularized FD, and the difference between Algorithm 1 and SCFD is still unclear. Hence, I decide to keep my rating.

---

> > > ### Author Response · Authors · 2023-08-21
> > > **Thanks for the feedback**
> > >
> > > Given the feedback, and the reviewer's note of similarity between SCFD, we wonder if the following presentation would be amenable to the reviewer?
> > >
> > > First, we can move algorithm 1 (S-Adagrad) to a section in related work, marking its similarity in the past literature to SCFD in the bandit context. The methods section already mostly focuses on our novel regret **analysis** of S-Adagrad / SCFD, which does not appear in prior work. It appears the reviewer is in agreement that S-Adagrad / SCFD has not yet been analyzed in the OCO setting with sqrt-T worst case regret?
> > >
> > > In this way, we can emphasize the main contribution of the non-Kronecker-factored case to be our analysis, rather than the algorithmic proposal. However, the key new algorithm unlocked by this OCO approach is through integration with Shampoo, which as our top comment demonstrates is critical for sublinear memory and thus application to DNNs.

---

> > > > ### Comment · Reviewer_fXUQ · 2023-08-22
> > > > **Thanks for your feedback**
> > > >
> > > > I believe it is more appropriate to reorganize the paper, emphasizing that the primary contribution lies in the analysis of the non-Kronecker-factored case rather than the algorithmic proposal. This suggests a major revision and an additional round of review are necessary.

---

### Official Review · Reviewer_rVs5 · 2023-07-06

**Soundness:** 3 good
**Presentation:** 2 fair
**Contribution:** 3 good
**Rating:** 6
**Confidence:** 4

**Summary:**

It is known that dense preconditioning can be highly beneficial for optimization, but this has high memory requirements and is often prohibitive. The classic solution to this problem is the use of a diagonal preconditioner. This work proposes the use of a low rank preconditioner by using a streaming approach to low rank approximation known as Frequent Directions, which maintains small memory requirements throughout training. The authors then give a regret bound on this algorithm, by combining the analysis of Frequent Directions with the regret analysis of AdaGrad, which results in a trade-off between the regret and the memory used for the preconditioner. While the idea of using Frequent Directions for compressing preconditioners in a stream has been used before, the authors obtain a bound that has advantages over prior regret bounds. Experimental results are nice.

**Strengths:**

This work gives a highly practical idea for efficient training of neural networks, with a nice theoretical analysis that combines AdaGrad and Frequent Directions as well as nice empirical results.

**Weaknesses:**

I am confused about the primary contribution of this work, and perhaps the writing needs to be improved to clarify this. The abstract and early introduction seem to mostly focus on the contribution of this work as the idea of applying Frequent Directions during optimization. This is confusing, as the idea itself has been suggested and investigated in prior work (which the authors thoroughly discuss).

**Questions:**

I still do not understand the difference between the current work and FD-SON, could you clarify?

**Limitations:**

Some discussion is provided in the Discussions section.

---

> ### Author Rebuttal · Authors · 2023-08-08
>
> Thank you for taking the time to read and review our work. Here are our responses to your questions and concerns, with which we hope to boost your confidence in our work:
>
> Weaknesses:
>
> **rVs5**: I am confused about the primary contribution of this work, and perhaps the writing needs to be improved to clarify this. The abstract and early introduction seem to mostly focus on the contribution of this work as the idea of applying Frequent Directions during optimization. This is confusing, as the idea itself has been suggested and investigated in prior work (which the authors thoroughly discuss).
>
> **Response**: Please see our top-level comment. In short, in L31, we emphasize the focus on the Kronecker-factored gradient covariance, not the full gradient covariance.
>
> Questions:
>
> 1. I still do not understand the difference between the current work and FD-SON, could you clarify?
>
> **Response**: We highlight Table 1, page 3 in our current work, where FD-SON regret simplifies to $\sqrt{\ell \lambda_{\ell:d} T}$ and our work has $\sqrt{d(d-\ell)\lambda_{\ell:d}}+tr(G_T)^{1/2}$. In the case that there is no spectral decay, so $\lambda_{\ell:d}=\Theta(T)$, the S-Adagrad analysis degrades gracefully, with still $\sqrt{T}$ regret. The FD-SON bound becomes linear in the OCO case with no spectral decay.
>
> We’ve updated the text to describe this case.

---

> > ### Comment · Reviewer_rVs5 · 2023-08-21
> > **Thank you for the response**
> >
> > Thank you for the clarifications. I did not realize the focus on the Kronecker-factored gradient covariance, although this doesn't change my initial impressions significantly. I will keep my score.

---

> > > ### Author Response · Authors · 2023-08-21
> > > **Please reconsider the focus on Kronecker factors.**
> > >
> > > We apologize if our earlier reply was too short to emphasize how the focus on Shampoo sketching and Kronecker factors results in dramatically different guarantees from FD-SON (please also note our preconditioner computation is quite different, in the matrix case requiring inverse fourth roots, rather than inverse matrices).
> > >
> > > A Kronecker-factored covariance results in $O(m^2+n^2)$ memory use for accumulators for an $m\times n$ tensor to optimize. With our sketching approach, this becomes $O(mk+nk)$, for sketching rank $k<\min(m,n)$, which is now sublinear. As our top-level comment emphasizes, this results in a dramatic memory decrease, asymptotically smaller compared to previous sketching approaches and linear-memory algorithms like Adam.
> > >
> > > Moreover, we provide novel analysis in the non-Kronecker-factored case (S-Adagrad), providing the first $O(\sqrt{T})$ bound for a sketched OCO algorithm in the literature, not present in past works such as FD-SON.
> > >
> > > Given two strong theoretical differentiations from FD-SON (1 - asymptotically less memory in the matrix case, 2 - square root regret bound for general OCO setting without curvature) and one practical one (our sublinear memory allows application to large deep neural networks, could the reviewer provide reasons for their reasons to reject, which they say outweighs limits to accept?
> > >
> > > In our top-level comment, we quote the contributions of our work from the introduction, which clarify the novelty in question.

---

> > > > ### Comment · Reviewer_rVs5 · 2023-08-21
> > > > **Thank you for the response**
> > > >
> > > > I appreciate the authors' efforts in defending their work, and I would like to better understand their points.
> > > >
> > > > My understanding of the work, taking the focus on the Kronecker product into account, is that it studies an FD-SON-like algorithm for matrix-shaped parameters (such as those for neural net weights), where instead of preconditioning the mn parameters as a vector, it preconditions the m x n matrix from the left and right. In this setting, it is shown that a low rank approximation of the left and right preconditioners gives a smaller memory algorithm with guarantees that interpolates correctly the full rank left/right matrix preconditioner case.
> > > > - Applying low rank approximation to asymptotically save on memory is an ubiquitous idea, and I do not think the community particularly benefits from having another paper that instantiates this idea. However, I would like to discuss the regret analysis more to better understand the contributions here.
> > > > - My initial impression was that analyzing regret bounds for Shampoo (without low rank approximation) would be straightforward and known in the literature. Is the claim that this is already a challenge that is newly contributed by this work? I believe this would be the point that makes the biggest difference in my impression of the work.
> > > > - Given an analysis of Shampoo, working the analysis of FD into the regret analysis, which I understood to be the main theoretical thrust of this work, seems like a marginal contribution to me. In particular, FD already gives the machinery to reason about how the escaped mass of low rank components evolve through streams of updates, and putting this into a regret analysis seems to me like just applying a theorem in a new context. Please let me know if there are any particular challenges that I should take another look at.

---

> > > > > ### Author Response · Authors · 2023-08-21
> > > > > **Thank you for continued interest & questions**
> > > > >
> > > > > We greatly appreciate the reviewer's interest in understanding the work! Thank you for your continued investment.
> > > > >
> > > > > Even in the vector-parameter case (S-Adagrad), low-rank second-moment algorithms have previously required either:
> > > > >
> > > > > 1. curvature (FD-SON and its derivatives do this) to yield an ONS-type algorithms
> > > > > 2. weak bounds (Radagrad is heuristic in low memory form, and see Observation 1 for Ada-FD).
> > > > >
> > > > > So even for analyzing the vector case, our Theorem 3 provides a novel worst-case sqrt-T guarantee when dealing with general convex functions with vector parameters. This is a gap in the literature and may be of interest due to recent work on lower bounds (Efficient Convex Optimization Requires Superlinear Memory by Marsden et al 2022).
> > > > >
> > > > > What we do differently from, say, Ada-FD, is set up a strictly more convex preconditioner, which allows us to establish the approximate isometry in Remark 9 in the Supplement (L569).
> > > > >
> > > > > This two-sided bound around the "ideal" full matrix preconditioner makes the FD preconditioner essentially a modular component in OCO proofs, so long as the algorithm applies the dominating approximation to an exact preconditioner: we rely on the same isometry for Shampoo's Kronecker factors $L_t,R_t$ (implied by Lemma 12). We hope the community can use the same technique in other OCO settings too; surely the community stands to gain from the demonstration of this technique in our work, as the OCO sqrt-T result was missing.
> > > > >
> > > > > Finally, we invite the reviewer to take a look at our empirical results, summarized in Figure 2. We believe that this another "challenge" which has been neglected in this line of work and not yet considered by the reviewer: empirical application of low memory algorithms to real optimization problems of interest to practictioners. As mentioned in Fig.1 and our top-level comment, existing low-memory approaches result in **superlinear** accumulator memory size. Perhaps on the theoretical side, low memory approaches are well known, but in practice algorithms such as RFD, SCFD, Ada-FD, Radagrad, GGT could not be applied to large networks such as the ResNet-50, Conformer, and GNN we have in our work. Storing multiple copies' worth of the parameters for these networks on GPUs is prohibitive.
> > > > >
> > > > > Adafactor may be the only real example of sublinear memory optimizers in practice, but it does not have a robust theory and performs much worse than Adam in terms of quality on all three of our datasets (notice we improve on Adam in all three).

---

> > > > > > ### Comment · Reviewer_rVs5 · 2023-08-21
> > > > > > **Thank you for the response**
> > > > > >
> > > > > > I appreciate the authors' in-depth discussion and clarification of their results, and for their patience with my last-minute questions on the last day of the rebuttal period.
> > > > > >
> > > > > > I still am not 100% convinced about the importance of the low rank-based optimizers in practice since it seems like a "low hanging fruit" idea, given my lack of experience in this area of practical implementations of optimizers and some preliminary searches, I trust that this is indeed a significant contribution.
> > > > > >
> > > > > > I am convinced that the theoretical results are more significant than I originally perceived, and I would like to raise my contribution score to a 3 (good) and the overall score to a 6 (weak accept), primarily on the basis of the theoretical results. I would prefer for the paper to be written more to focus on the discussion of the theoretical results, although this is perhaps a more personal taste and the other reviewers may disagree.
> > > > > >
> > > > > > Thank you again to the authors for their patience.

---

> > > > > > > ### Author Response · Authors · 2023-08-21
> > > > > > > **Thank you for the review!**
> > > > > > >
> > > > > > > We really appreciate the reviewer's questions and their engagement on our work. This is a great experience for us as authors and, based on the feedback, helps us modify text to direct attention in the methods section to the construction of the reusable concepts (such as the approx isometry) in our proofs!

---

### Author Rebuttal · Authors · 2023-08-08

Thank you for all the reviewers for taking the time to read and review our work. Many reviewer comments focus on the differences between S-Adagrad and prior work which inspects FD for optimization.

We invite the reviewers to look at the following sections of our paper. As mentioned in L31, the Kroneker-factored gradient is of particular interest because FD applied to each factor results in sublinear usage. We discuss this in:

The first of our contributions L37-L41 “we recover full-matrix AdaGrad regret up to additive spectral terms under a memory constraint, providing a novel guarantee without curvature assumptions” (not present in the literature) and “[r]igorously composing our approach with Shampoo (Sec. 4.2), unlocks a second order algorithm which requires sub-linear memory for its accumulators.”
Related work L62-68: “approximating the factored covariance $L_t\otimes R_t$ requires less memory than the full covariance and explains why our method can scale to large modern architectures whereas Agarwal et al. [6] cannot”. Note this same explanation applies to RFD,  SCFD, Ada-FD, Radagrad, etc.
Figure 1 on page 4, which diagrams how historical sketch-based methods require superlinear memory $O(mnr)$ for a matrix size $m\times n$ and Sketchy requires $O(mk + nk)$, which is sublinear.

We attempted to emphasize the two differences (novel guarantee, sub-linear memory) from prior work by bolding them in the introduction in the contributions portion at the end of the introduction, but we can include additional discussion in the first page as well.

Many of these historical low-memory approaches have failed to gain traction in the community. We believe that because they did not have $\sqrt{T}$ regret in the OCO case, composition with Shampoo was never a natural option for these methods. Since they all require essentially multiple copies of the neural network weights to be stored to operate, they were never of much interest to practitioners (whose expectations are based on Adam, which requires just one copy for accumulators).

Several reviewers have asked about wall-clock time. We emphasize that we did not perform extensive engineering optimizations. Both Sketchy and Shampoo have iterative variants, but SVD/eigh were used, respectively, for implementation simplicity. For all datasets, numbers below represent the 95% bootstrap confidence interval constructed from taking the median of steps/sec measurements across 5 runs per dataset/algorithm pair. Each run had about 50 such measurements taken uniformly over the course of training. Librispeech data was unfortunately unusable as the cluster scheduler we used interrupted execution frequently, invalidating measurements.

Steps per second (higher is better), for setup described in appendix / Figure 2

**Imagenet**

   Sketchy [6.173, 6.348]

   Adam [14.909, 18.904]

   Shampoo [2.956, 3.032]

**OGBG-MolPCBA**

   Sketchy [4.193, 4.238]

   Adam [4.426, 4.526]

   Shampoo [3.522, 3.766]

Even without heavy engineering, Sketchy improved in runtime over Shampoo in this replicated data-parallel setting (16 TPUv4 per run) because even without an iterative routine, it performs a reduced-size SVD (per L287 in the paper) instead of a full decomposition for gradient KF factors.

---

### Decision · Program_Chairs · 2023-09-21

**Decision:**

Accept (poster)

**Comment:**

Dear Authors,

Thank you for your valuable contribution to Neurips and the ML community. Your submitted paper has undergone a rigorous review process, and I have carefully read feedback provided by the reviewers and considered the author rebuttal in detail.

This paper proposes the use of low rank preconditioners in optimization via the Frequent Directions (FD) method, which is a streaming low-rank approximation algorithm maintaining a low memory budget. The authors provide a theoretical regret bound and present numerical simulations.

While all the reviewers agree that the work is interesting and potentially impactful, the paper received some critical reviews. In particular, Reviewer fXUQ was initially highly critical regarding the novelty of the method. The reviewer pointed out connections with existing works that apply FD in optimization. The author rebuttal addressed some of these comments and clarified that the main contribution is in the analysis of the non-Kronecker-factored case. The reviewer argues that a reorganization is needed to clarify the contributions and connections to existing proposals (e.g., Spectral Compensation Frequent Directions). Although the theoretical analysis appears to be a definite improvement, in the reviewer discussion phase, it was not entirely clear how much of the algorithmic proposal is novel. In addition, Reviewer 6Jc9 was critical about the theory and experiments. The reviewer claimed that the theoretical bound is loose and the experiments are not conclusive. The author rebuttal led the reviewer to increase their score. We congratulate the authors for the effective rebuttal, which clarified some concerns.

I am pleased to recommend the acceptance of your paper for publication at our conference. This is an excellent achievement, and I congratulate you on your success. As we move forward with the publication process, I would like to remind you to carefully review the feedback and comments provided by the reviewers. While the overall assessment is positive, the reviewers have offered valuable suggestions that can further strengthen the quality of the paper.

In particular, as Reviewer fXUQ suggested, I strongly urge the authors to discuss connections with Spectral Compensation Frequent Directions (SCFD) and make sure the contributions are not overstated.

I encourage you to use this feedback to make any necessary improvements and refinements to the manuscript before submitting the final version for publication.

Once again, thank you for submitting your work to our conference, and congratulations on this well-deserved recognition. I look forward to seeing your paper published in our conference proceedings and witnessing the impact of your research on the scientific community.

Best,
AC